# DNA Specimen Preservation Using DESS and DNA Extraction in Museum Collections

**DOI:** 10.3390/biology14060730

**Published:** 2025-06-19

**Authors:** Eri Ogiso-Tanaka, Daisuke Shimada, Akito Ogawa, Genki Ishiyama, Ken-ichi Okumura, Kentaro Hosaka, Chikako Ishii, Kyung-Ok Nam, Masakazu Hoshino, Shuhei Nomura, Showtaro Kakizoe, Yasuhide Nakamura, Isao Nishiumi, Minako Abe Ito, Taiju Kitayama, Norio Tanaka, Tsuyoshi Hosoya, Utsugi Jinbo

**Affiliations:** 1Center for Molecular Biodiversity Research, National Museum of Nature and Science (NMNS Tokyo), Amakubo 4-1-1, Tsukuba 305-0005, Ibaraki, Japannamakogawa@kahaku.go.jp (A.O.); gishiyama@kahaku.go.jp (G.I.); khosaka@kahaku.go.jp (K.H.); ujinbo@kahaku.go.jp (U.J.); 2Institute for Extra-Cutting-Edge Science and Technology Avant-Garde Research (X-STAR), Japan Agency for Marine-Earth Science and Technology (JAMSTEC), Natsushima-cho 2-15, Yokosuka 237-0061, Kanagawa, Japan; 3Department of Zoology, National Museum of Nature and Science (NMNS Tokyo), Amakubo 4-1-1, Tsukuba 305-0005, Ibaraki, Japan; okumura@kahaku.go.jp (K.-i.O.); nomura@kahaku.go.jp (S.N.); nishiumi@kahaku.go.jp (I.N.); 4Department of Botany, National Museum of Nature and Science (NMNS Tokyo), Amakubo 4-1-1, Tsukuba 305-0005, Ibaraki, Japanjasnakamura@soc.shimane-u.ac.jp (Y.N.); kitayama@kahaku.go.jp (T.K.); ntanaka@kahaku.go.jp (N.T.); hosoya@kahaku.go.jp (T.H.); 5Research Center for Inland Seas, Kobe University, Rokkodai 1-1 Nadaku, Kobe 657-8501, Hyōgo, Japan; mhoshino.sci@gmail.com; 6Center for Collections, National Museum of Nature and Science (NMNS Tokyo), Amakubo 4-1-1, Tsukuba 305-0005, Ibaraki, Japan; kakizoe@kahaku.go.jp (S.K.); ito-minako@kahaku.go.jp (M.A.I.); 7Estuary Research Center, Shimane University, 1060 Nishikawatsu-Cho, Matsue 690-8504, Shimane, Japan

**Keywords:** DNA, specimen preserved methods, nematodes, invertebrates, birds, fungi, seagrasses, algae, museum collection

## Abstract

Herein, we investigated the effectiveness of traditional preservation methods (such as ethanol) versus DESS (DMSO/EDTA/saturated NaCl solution) with regard to maintaining both morphology and DNA integrity using our museum specimens. Our results demonstrated that preserving tissues in DESS maintained high molecular weight DNA across all examined species at room temperature. When preserving whole organisms, the optimal preservation solution conditions for maintaining both morphological features and DNA integrity varied among species, but DESS was effective for preserving both morphology and DNA except for species with calcium carbonate skeletons and shells. These findings suggest that DESS utilization for specimen DNA preservation is effective across many species, not only for long-term storage in environments without freezer facilities but also for temporary preservation until freezing is possible.

## 1. Introduction

Biological specimens preserved in museums’ collections constitute tangible records of natural states from historical timepoints, serving as repositories of information that may be useful in future discoveries [1]. Such specimens enable researchers to elucidate historical climate fluctuations, environmental conditions, and ecosystem dynamics. Recent technological advancements have facilitated the extraction and analysis of residual DNA from whole specimens or their fragments. This ongoing technological progress continues to unlock new information from museum collections.

The systematic approach to extracting new information from museum specimens was termed “Museomics” (a portmanteau of “museum” and “-omics”) by Dr. Stephan Schuster in 2007 [2]. Museomics is a multidisciplinary field dedicated to retrieving biochemical and molecular biological data from museum collections. Dr. Schuster and his group successfully reconstructed substantial portions of the mammoth genome by applying next-generation sequencing technology to mammoth specimens that had been preserved in a museum for 200 years [3]. Furthermore, multiple research groups have successfully extracted DNA from specimens that were over a century old and implemented DNA barcoding techniques [4] for biodiversity studies and sequence acquisition across diverse taxonomic groups [5,6,7,8,9]. While Museomics research yields valuable insights by extracting comprehensive information from specimens, the analytical processes frequently necessitate the utilization of specimen fragments, resulting in the gradual deterioration of these irreplaceable resources. Consequently, there is an imperative need for the development of nondestructive analytical methodologies and techniques for the stable, long-term preservation of specimen information.

DNA has emerged as a powerful tool for taxonomic identification, occasionally being preserved through the cryopreservation of tissue samples linked to voucher specimens; however, implementing this approach universally across all specimen collections presents significant challenges. Moreover, institutions lacking cryogenic facilities or those managing microorganisms and small invertebrates require methodologies that simultaneously maintain the morphological integrity of the specimen while ensuring stable preservation of its DNA. Conventional standard preservation reagents, which include anhydrous ethanol and RNA later Stabilization Solution (Thermo Fisher Scientific, Waltham, MA, USA) are currently employed to inhibit DNA degradation during biological sample storage [10,11]. However, these solutions induce tissue dehydration, potentially compromising morphology integrity, thus limiting their efficacy as optimal preservatives for taxonomic purposes [12]. Furthermore, while so-called nondestructive DNA extraction methods have been developed for insects and plants, these approaches typically utilize proteinase K, making it difficult to classify them as truly nondestructive methodologies [13,14]. Organisms lacking exoskeletons or cell walls are especially vulnerable to degradation via proteinase K, with the potential of complete dissolution and elimination of the specimen.

DESS (dimethyl sulfoxide, ethylenediamine tetra-acetic acid, and saturated salt) solution is a saturated NaCl solution containing 20% dimethyl sulfoxide (DMSO) and 250 mM ethylenediamine tetra-acetic acid (EDTA) that is widely utilized for DNA preservation in biological tissue samples. DESS was initially developed for preserving avian tissue DNA at room temperature (RT) [15]. Subsequent research demonstrated the efficacy of DESS in preserving both morphological structures and high molecular weight DNA in various invertebrates [16], including marine, terrestrial, and parasitic nematodes [17]. Recent molecular biological investigations have further validated the maintenance of high molecular weight DNA integrity in DESS under ambient temperature conditions [18,19].

Our previous investigations demonstrated the successful extraction of DNA from the supernatant of nematode specimens preserved in DESS at RT for a decade, enabling effective DNA barcoding [20]. Most free-living nematodes have small body dimensions (0.1–5.0 mm) and lack detachable appendages, which frequently results in complete specimen consumption during conventional DNA extraction procedures, thereby precluding specimen retention. However, nondestructive expression methodologies present opportunities for maintaining intact specimens throughout molecular analysis. Therefore, the present study investigated the characteristics of the DESS preservation solution and evaluated its applicability across diverse biological specimens from the National Museum of Nature and Science (Tokyo, Japan) collections, assessing its effects on morphological preservation, DNA stability, and efficacy for nondestructive DNA barcoding applications.

## 2. Materials and Methods

### 2.1. Materials

#### 2.1.1. Specimens 

Nematode: All nematode specimens used in this study were free-living marine nematodes collected from Japan between 2012 and 2019 and in 2022 [20,21,22] (Table 1). For specimen preparation methods, refer to [20].

Arthropoda Insecta: *Prinus insularis* was collected on 30 June 2024, in the parking lot of the NMNS, Japan (Table 1). The legs were separated into the femur and tibia. The tibiae were preserved under six different conditions: (1) anhydrous ethanol; (2) 70% ethanol; (3) mixture of 70% ethanol and DESS (19:1); (4) 25% DESS; (5) 50% DESS; and (6) 90% DESS. The femora were preserved under six different conditions: (7) air-dried after 24 h immersion in ethanol; (8) air-dried after 24 h immersion in condition 3; (9) solution containing 44% ethanol, 0.5 M NaCl, 8% DMSO, and 200 mM EDTA; (10) 5% DESS; (11) 10% DESS; and (12) air-dried after 24 h immersion in condition 6. All samples were stored at room temperature (15–30 °C) in a laboratory drawer under dark conditions. After 5.5 months, excess moisture was removed from each sample using paper towels before DNA extraction. A total of 20 specimens from various families of Coleoptera and Membracidae were collected on 30 June 2022 in Kiso Village, Nagano, Japan, using a light trap. Those specimens were placed directly into 100% DESS in several tubes. *Protaetia orientalis* was collected at the NMNS botanical garden on 4 August 2023 and placed directly into the 100% DESS solution in alive condition. The wings of *P. orientalis* preserved in 40 mL DESS for one and a half years were divided into two parts (resulting in four wing sections) and then immersed in either water or 70% ethanol at a ratio of 100-fold the wing weight (approximately 2 mL) for 3 min to remove DESS solution components. The wings were then placed on paper towels to thoroughly remove excess moisture and air-dried for 1 h. After air-drying, two of the four wings were brushed with a paintbrush moistened with 70% ethanol to remove surface contaminants. All four wings were further air-dried for 2 h before examining their surface using a Scanning Electron Microscope (SEM).

Spider: Seventy-seven specimens of 35 species were used, which were collected from all over the country between 2016 and 2023 (Table 1 and Appendix A). All specimens were stored in liquid. The specimens collected in 2023 were mainly stored by DESS, separated into multiple concentrations. Other than these, anhydrous ethanol (99.5%) or 75% ethanol was used for storage. In some specimens, only the leg was preserved for data extraction.

Neomysis: *Neomysis japonica* was collected at Lake Shinji, Japan, on 29 September 2022 (Table 1). The specimen was immediately preserved in 1.5 mL of 100% DESS, 99% ethanol, and 5% formalin. After 7 days, 700 µL of DESS supernatant was removed, and an equal volume of water was added to dilute the DESS concentration to 50% on 5 October 2022. The specimens were then stored at room temperature in a laboratory drawer for 18 days. Subsequently, 700 µL of supernatant was collected for DNA extraction. The supernatant was replaced with 100% DESS, and the specimens were preserved under the same conditions. The specimens were continuously preserved, and microscopic observations were conducted on 15 December 2024.

Holothuroidea, Echinodermata: Three species of holothurians (Table 1) were collected during the KH-22-8 cruise of the R/V Hakuho Maru (Japan Agency for Marine-Earth Science and Technology and Atmosphere, JAMSTEC and Ocean Research Institute, the University of Tokyo, AORI) using bottom contacted beam trawls. *Pseudostichopus* sp. was collected from the Kuril-Kamchatka Trench, south of Hokkaido, Japan (41°20.786′ N to 41°20.517′ N, 144°56.455′ E to 144°57.297′ E: St. A3 BT) from 6116 to 6131 m, on 3 October 2022. *Peniagone* sp. was collected from Kuril-Kamchatka Trench, south of Hokkaido, Japan (41°11.786′ N to 41°12.354′ N, 144°43.042′E to 144°43.439′ E: St. A4b BT) from 6799 to 6807 m, on 6 October 2022. *Elpidia kurilensis* was collected from the Japan Trench, east of Miyagi, Japan (38°33.673′ N to 38°32.980′ N, 144°05.371′ E to 144°05.381′ E: St. G BT) from 7651 to 7654 m, on 15 October 2022. Holothurian specimens were preserved under two different solutions at room temperature in a laboratory: (1) whole bodies were preserved in 40 mL of 100% DESS; (2) whole bodies were preserved in 99.5% ethanol. As the control for DNA fragmentation was affected by different preservation solutions, a piece of another specimen of *Peniagone* sp. was collected during the same cruise and preserved in ethanol for about 40 days and stored at −20 °C.

Gastropoda: *Lottia kogamogai* and *Echinolittorina radiata* were collected at the intertidal zone of the Sasago Coast, Shimane Prefecture, Japan, on 16 September 2024 (Table 1). The specimens were immersed in 100% DESS. Two specimens of *Nassarius siquijorensis* were sampled at Ise Bay by dredge of the Research Vessel Seisui-Maru and preserved in 100% ethanol. Four specimens of *Sinotaia quadrata histrica* were sampled at Kasumigaura, Ibaraki Prefecture, Japan on 28 December 2024. They were boiled in 80–90 °C water. Each soft body was preserved in the following ways, respectively: (1) with shell in 100% DESS, (2) without shell in 100% DESS, (3) with shell in 100% ethanol, and (4) without shell in 100% ethanol. All samples were stored at room temperature in a laboratory.

Birds: Chicken breast meat purchased from a supermarket was used as a model sample (Table 1). The peripheral tissues of the breast meat that had been exposed to air were not used; instead, muscle tissue from the central part of the breast was excised using surgical scissors. Three species of muscle tissue (*Otus lettia*, *Pellorneum ruficeps*, *Sturnia malabarica*) were preserved in 100% DESS at RT under dark conditions for 2 years. Parts of the muscle tissue in the preservative solution, exfoliated tissue, and tissue adhering to the tube wall were used.

Seagrasses: *Zostera marina* and *Phyllospadix iwatensis* were collected from Denshinhama, Muroran, Hokkaido, on 18 August 2023 (Table 1). The samples were immediately placed in plastic bags and kept cool in a cooler box. After 4 days, they were transferred to 50 mL tubes filled with DESS solution and stored at 4 °C. After 7 months of storage, DNA was extracted from 10 cm leaf segments. The remaining samples in DESS were stored at room temperature on a laboratory bench (light/dark conditions). After 16 months of storage, the same DNA extraction procedure was repeated. As a control, the samples collected and processed using our standard method were used (see Section 2.2.2. DNA extraction), Method 2. These control samples were kept cool in a cooler box after sampling, dried with silica gel, and then ground for DNA extraction. *Z. marina* was collected from Utoro, Hokkaido, on 29 August 2023, with DNA extracted on 1 October, while *P. iwatensis* was collected from Muroran, Hokkaido, on 18 August 2023, with DNA extracted on 29 August 2023.

Algae: A brown alga, *Turbinaria ornata* (Turner) J.Agardh (Fucales, Phaeophyceae) was collected at the tide pool on the beach, Otchogahama, Hachijo Island, Izu Islands, Tokyo Metr., Japan on 16 September 2024 (Table 1). The specimen was immediately immersed in 20 mL of DESS, transported to the laboratory, and stored at room temperature.

Fungi: A total of 6 mushroom species (2 orders, 4 families, and 5 genera in the phylum Basidiomycota) were collected on February 2010 (Table 1) and preserved in a modified DESS (original DESS with Tris and Na_2_SO_3_ as described by Hosaka and Castellano, 2008 [23]; hereafter referred to as DESS-NMNS). The samples were stored at room temperature until the initial DNA extraction was conducted in March 2010, and then at 4 °C until now. The voucher specimens of each sample were prepared as air-dried specimens as described by Hosaka and Uno (2011) [24] and stored at the fungal herbarium of the National Museum of Nature and Science (TNS).

#### 2.1.2. DESS and DESS-NMNS Solution

Two preservation solutions, DESS [17] and its modified version [23], were prepared for DNA and tissue preservation. Their compositions are shown in Table 2.

A total of 500 mL of DESS was prepared according to the following procedure: 46.53 g of EDTA was placed in a 500 mL glass beaker with 200 mL of distilled water, and 10 M NaOH was added to adjust the pH to 8.0. After pH adjustment, the EDTA solution was brought to 250 mL with distilled water using a measuring cylinder. A total of 100 mL of DMSO and 150 mL of distilled water were added to a new 1 L glass beaker, followed by the addition of 250 mL EDTA solution from the measuring cylinder. Approximately 100 g of NaCl was added to the beaker until saturation and mixed overnight at room temperature (about 20 °C) using a magnetic stirrer. The supernatant was transferred to a new 500 mL bottle. The bottle must be stored under dark conditions. For 500 mL DESS-NMNS preparation, the final 150 mL of distilled water was replaced with 50 mL of 1 M Tris-HCl (pH 8.0), 75 mL of distilled water, and 25 mL of filtered, autoclaved Na_2_SO_3_.

### 2.2. Methods

#### 2.2.1. Analysis of Evaporation Rates in Preservation Solutions

Evaporation rates were measured by monitoring the weight changes in water (as control), ethanol and DESS stored in 1.5 mL safe-lock tubes (Cat No. LT-0150, BIO-BIK Ina Optica, Osaka, Japan) and 2 mL screw cap tubes (Cat No. T330-6, Tokyo Garasu Kikai Co., Ltd., Tokyo, Japan) using an analytical balance over time. The 2 mL screw cap tubes used in this study are identical to those utilized in our museum (NMNS, Tokyo, Japan).

#### 2.2.2. DNA Extraction

Method 1: A modified protocol of DNeasy Blood and Tissue Kits for DNA Isolation (QIAGEN Cat. No 69504) was used. A part of each specimen (insect, spider legs, whole nematodes, a piece of bird muscle, a tube feet or body wall of holothurians, and a foot muscle) was placed in a 1.5 mL LoBind tube (Eppendorf, Hamburg, Germany, Cat. No. 0030108051). The sample was mixed with 180 µL of ATL buffer (QIAGEN, Hilden, Germany, Cat. No. 939011) and 10~20 µL Proteinase K (QIAGEN, Cat. No. 19134) by mixing, then incubated at 55 °C from 6 h to overnight. After incubation, 200 µL AL buffer (QIAGEN, Cat. No. 19075) was added and mixed thoroughly by vortexing. If the solution became turbid, the mixture was heated at 70 °C for 10 min until it became clear before proceeding the next step. The solution was thoroughly mixed by vortexing after adding 210 µL of ethanol. After mixing well, 10 µL of silica suspension was added and mixed for at least 1 min. The mixture was centrifuged at 10,000 rpm for 1 min, and the supernatant was discarded. The pellet was washed by adding 300 µL of wash buffer, followed by vortexing. After centrifugation at 10,000 rpm for 1 min, the supernatant was discarded. This washing step was repeated once more with wash buffer. After completely removing the final supernatant, the pellet was air-dried in dark conditions, taking care not to over-dry the silica. Finally, DNA was eluted by adding 50 µL of 0.1× TE buffer. The composition and preparation of silica suspension and wash buffer are described in [20].

Method 2: A modified protocol of the CTAB method [25] was used. A part of the Seagrasses (approximately 20 mg dry weight), specifically clean portions from leaf tips or near-root regions without visible epiohytes, was finely cut with tweezers, placed in a 2 mL tube with disruption beads (one 5 mm zirconia bead, three to four 3 mm zirconia beads, and approximately one spatula of 1 mm zirconia beads), and homogenized using a Tissuelyser III at 27 Hz for 2 min (QIAGEN). A total of 500 µL of CTAB buffer [2xCTAB buffer: Lysis buffer: 2-mercaptoethanol = 10:1:0.1] was added, and the mixture was incubated at 60 °C for 30 min with occasional mixing. A total of 500 µL of chloroform–isoamyl alcohol (CI = 24:1) was added and thoroughly mixed by vortexing, and centrifuged at 15,000 rpm for 5 min at 20 °C. The supernatant was transferred to a new 1.5 mL tube, and 425 µL of isopropanol was added and mixed gently. The tube was placed in a centrifuge at 4°C for approximately 5 min, then centrifuged at 12,500 rpm for 7 min. The pellet was retained and 500 µL of 70% ethanol was added, followed by centrifugation at 12,500 rpm for 7 min at 4 °C. The supernatant was discarded leaving the pellet, which was then air-dried. The DNA was eluted in 50–100 µL of TE buffer. The composition of each buffer was as follows: 2xCTAB Buffer was prepared by dissolving 10 g CTAB, 41 g NaCl, 50 mL of 1 M Tris-HCl (pH8.0), and 3.7 g of EDTA-2Na in water and making up to 500 mL. Lysis Buffer contained 10 g of N-lauryl Sarcosine Sodium Salt, 10 mL of 1 M Tris-HCl (pH 8.0), and 4 mL of 500 mM EDTA.

Method 3: A modified protocol of [26,27] was used. A part of the specimen (Seagrasses and Algae) in DESS was placed in a pre-chilled mortar on ice containing 1 mL of Lysis buffer and thoroughly ground. After confirming complete homogenization, 5 mL of Lysis buffer was added and mixed well, then transferred to a 50 mL tube and incubated in a 65 °C water bath with constant agitation for 2 h. The mixture was then shaken at room temperature for 2 days. After centrifugation at 10,000 rpm for 3 min, 5 mL of the supernatant was transferred to a new 50 mL tube. 10 mL of Binding buffer was added and thoroughly mixed by vortexing, followed by centrifugation at 10,000 rpm for 3 min. The supernatant was removed by aspiration, and 5 mL of wash buffer 1 was added to the diatomaceous earth pellet in the 50 mL tube, mixed well, and centrifuged at 10,000 rpm for 3 min before removing the supernatant. A total of 15 mL of wash buffer 2 was added, mixed well, and centrifuged at 10,000 rpm for 3 min, with the supernatant removal step repeated twice. Subsequently, the diatomaceous earth pellet with a small amount of remaining supernatant was transferred to a Lobind 2 mL tube (Eppendorf, Germany, Cat. No.0030108078), centrifuged at 10,000 rpm for 3 min, and the supernatant was completely removed before air-drying. DNA was eluted by adding 100 µL of 0.1× TE and incubating at 55 °C for 10 min.

Method 4: A modified protocol of OmniPrep^TM^ for Plant (G-Biosciences Cat. No. 786-397) was used. A part of the thallus (approximately 76 mg wet weight) was finely cut with a tweezer, placed in a 2 mL tube, and frozen in liquid nitrogen. The frozen tissue was then homogenized using SK MILL (Tokken, Inc., Kashiwa, Japan) in the tube. After confirming complete homogenization, 500 µL of Genome Lysis Buffer (G-Biosciences, St. Louis, MO, USA) was added with 5 µL of Proteinase K (G-Biosciences), mixed by inverting the tube, and incubated at 60 °C for 2 h with occasional mixing. After cooling the sample to room temperature, 200 µL of chloroform was added and mixed by inverting, followed by centrifugation at 14,000× *g* for 10 min. The upper phase was transferred to a clean 1.5 mL tube, 50 µL of DNA Stripping Solution (G-Biosciences) was added, mixed by inverting, and incubated at 60 °C for 10 min. Then, 150 µL of the Precipitation Solution (G-Biosciences) was added, and mixed by inverting until a white precipitate was produced. The sample was incubated on ice for 15 min, centrifuged at 14,000× *g* for 10 min, and the supernatant was transferred to a clean 2 mL tube. To precipitate genomic DNA, 500 µL of isopropanol was added to the tube. The tube was mixed by inverting, centrifuged at 14,000× *g* for 10 min, and the supernatant was carefully removed. To wash the DNA pellet, 700 µL of 70% ethanol was added to the tube and centrifuged at 14,000× *g* for 10 min. After removing the ethanol, the tube was inverted on a clean absorbent surface for 5 min to allow excess ethanol to drain away. DNA was eluted by adding 50 µL of TE Buffer (G-Biosciences) with 0.5 µL of LongLife^TM^ RNase (G-Biosciences, Cat. No. 786-040) and incubating at 60 °C for 10 min.

Method 5: DNA was extracted from the tissue fragments stored in DESS-NMNS buffer as described above. Tissues were ground under liquid nitrogen conditions using a mortar and pestle. DNA extractions used a modified CTAB extraction followed by glass milk purification methods as detailed by [28]. The samples were incubated at 65 °C for 1 h and centrifuged at 12,000 rpm for 5 min. Only the aqueous phase was transferred to a new tube, and precipitated tissue debris was discarded. An equal volume of the mixture of chloroform–isoamyl alcohol (24:1) was added to the buffer, mixed vigorously for 2 min, and centrifuged at 12,000 rpm for 15 min at room temperature. The aqueous phase was pipetted out and transferred to a new tube. This step, using the chloroform/isoamyl alcohol mixture, was conducted only once.

After transferring ca. 300 μL of the aqueous phase, 1000 µL of 6.5 M sodium iodide buffer (6.5 M NaI, 50 mM Tris–HCl (pH 6.5), 10 mM EDTA, and 0.1 M Na_2_SO_3_) was added and mixed gently for 1 min. The silica mixture was prepared following the protocol of Rogstad (2003) [29], and 25 µL of the mixture was added to the samples. The samples were incubated at room temperature for 1 h and centrifuged at full speed for ca. 10 s. The supernatant was discarded and 1000 µL of washing buffer (10 mL Tris–HCl (pH 7.4), 1 mM EDTA, 100 mM NaCl, and 50% EtOH) was added, mixed briefly, and centrifuged at full speed for ca. 5 s. This washing step was repeated twice. After finishing the washing step, the samples were centrifuged one more time at full speed for 10 s. The remaining washing buffer was pipetted out, and the precipitated silica was dried at room temperature for 5 min.

The final elution was performed by adding 100 µL of ultra-pure water, followed by a brief mixing, and incubation at room temperature for 15 min. The samples were centrifuged at 12,000 rpm for 1 min, and the supernatant layer was transferred to a new tube and stored at −20 °C until PCR was performed.

#### 2.2.3. Assessment of DNA Quantification and Quality

Genomic DNA was quantified using the Qubit^TM^ dsDNA High Sensitivity or Broad Range Assay with the Qubit^TM^ 2.0 fluorometer (Thermo Fisher Scientific, USA). The fragment length distribution and DIN (DNA Integrity Number) were analyzed using the TapeStation 4200 system (Agilent Technologies, Inc., Santa Clara, CA, USA) with the genomic DNA ScreenTape Assay (Agilent).

#### 2.2.4. PCR for DNA Barcoding Region

COI: PCR was performed using COI universal primers (LCO1490/HCO2198, [30]) or (mlCOIint/HCO2198 [31]). The stock primers (100 μM) were diluted with a 0.1× TE buffer to prepare primer mixes: one containing 2 μM each of LCO1490 and HCO2198, and another containing 10 μM of mlCOIint, and 2 μM of HCO2198. PCR mixtures (20 μL final volume) contained 1 μL of DNA, 1 μL of primer pair, 10 μL of 2× Gflex Buffer, 0.2 μL of Tks Gflex DNA Polymerase (Cat. No. R060A, Takara, Maebashi, Japan), and 6.8 μL of ddH_2_O. Cycling conditions for COI were as follows: one cycle at 94 °C for 2 min, followed by 40 cycles at 94 °C for 30 s, 48 °C for 30 s, and 72 °C for 1 min (30 s in mlCOIint/HCO2198 primer pair), and final extension at 72 °C for 1 min.

Ribosomal tandem repeat (18S-ITS-26S): For amplifying the ITS region (ca. 600 bp.), a general primer pair of ITS5 and ITS4 [32] was used. PCR reactions were carried out using 10 μL reaction volumes each containing: 1 μL genomic DNA, 0.25 μL of each primer (10 μM), 5 μL of EmeraldAmp MAX PCR Master Mix (TaKaRa, Shiga, Japan), and 3.5 μL ultrapure water. PCR reactions were performed using the following parameters: 95 °C for 3 min; 35 cycles of 95 °C for 1 min, 51 °C for 45 s, 72 °C for 1 min; and 72 °C for 15 min. PCR products were electrophoresed in 1% agarose gels, stained with ethidium bromide, and visualized under UV light. To assess the quality (fragmentation and quantity) of extracted DNA, longer regions of the same ribosomal tandem repeat were also amplified. For this purpose, the primer pair of NS1/LR7 (ca. 5300 bp.) and NS1/LR12 (ca. 7100 bp.) [33] were amplified with the same PCR cycle parameters, except for a longer extension time of 2 min.

## 3. Results and Discussion

### 3.1. Characteristics of the DESS Solution

#### 3.1.1. The pH Stability of the DESS Solution

For DNA stability, the preservation solution must maintain slightly alkaline conditions (pH 7.5–8.0). Although we confirmed that DESS can preserve DNA and nematode morphology for at least 10 years in our previous study [20], it remains unclear how well DESS maintains its pH during long-term specimen storage. Although the DESS preservative solution contains EDTA with a weak pH buffering capacity, it lacks strong pH buffering substances such as Tris, raising concerns about pH decreases due to the dissolution of atmospheric carbon dioxide (CO_2_). When we investigated the pH of DESS (pH 7.5 and 8.0) that had stored nematodes for 10 years, although overall the pH values showed a decrease of 0.1 to 0.5 pH units, there was no observable trend of pH decrease with storage time (Figure 1a). This result was likely due to minimal air contact, as the tubes were filled almost to the top with DESS and were rarely opened during the 10-year period. Despite the fact that polypropylene tubes sometimes permit CO_2_ permeation [34], the absence of a pH decline over time was likely attributable to the extremely high concentration (250 mM) of EDTA in the DESS. From these results, a pH decrease in the DESS is unlikely to be a concern over several decades. However, for samples requiring multiple tube openings, tubes with imperfect seals, or to ensure longer-term pH stability, it may be desirable to add pH buffering agents such as Tris-HCl. Indeed, Hosaka et al. developed a modified version of DESS (referred to here as DESS-NMNS) that included Tris-HCl [23]. When the pH of the DESS-NMNS in liquid-preserved specimens of fungal fruiting bodies (i.e., mushrooms) sampled at RT over a period of 14 years was measured, all samples maintained pH levels above 7.76 (Figure 1b). Furthermore, no significant differences were observed between the samples with or without Tris addition over the approximately 10-year period. Because experimental tubes are typically made of polypropylene, which allows minimal permeation of O_2_ and CO_2_, further long-term observation is needed, including consideration of the durability of the polypropylene.

The cellular pH of most eukaryotic organisms is near neutral [35]. When organisms or their tissues are preserved in DESS, cellular components can cause a decrease in the pH of the DESS. As shown in Figure 1, nematodes cause a pH decrease of 0.1 to 0.5 units, suggesting that DESS with a pH of 8.0 might be more suitable than pH 7.5 DESS for the long-term storage of nematodes. We investigated the pH changes in various organisms, including sea cucumbers (Holothuroidea), seagrass, insects, bird muscle, algae, and mollusca. When a whole sea cucumber specimen was preserved in 50 mL of DESS at pH 8.2 and RT, the pH showed values of 7.6–7.8, indicating a decrease of 0.4–0.6 units (Figure 2a). Under similar conditions, when seagrasses were preserved in DESS, seagrass (*Zostera marina*) showed a pH decrease of 0.4–7.8, while *Phyllospadix japonica* showed a minimal pH change. This difference is likely due to the presence of sulfates in *Z. marina* [36]. The pH measurements taken after 2 years (sea cucumbers) or 1 year (seagrasses) of preservation showed little change. For samples preserved in DESS at pH 8.05, the insects showed a pH decrease of approximately 0.2–0.3 units (Figure 2b), while bird muscle tissue showed minimal change with a decrease of less than 0.1 units. Brown algae (*Turbinaria ornata*) exhibited a pH decrease of 0.2 units. In contrast, all molluskan specimens showed increasing pH values. In particular, specimens preserved with intact shells (Mollusca 1–3 in Figure 2b) showed increases of 0.5–1.2 units, likely due to the dissolution of calcium carbonate from the shells. Mollusks 1 (*Lottia kogamogai*) and 3 (*Sinotaia quadrata histrica*) were over 1 cm in size with substantial shells, while Mollusk 2 (*Echinolittorina radiata*) was approximately 3 mm in size, suggesting that the shell size correlates with the rate of pH increase. Although Mollusk 4 (*L. kogamogai*), with an absent shell, showed the lowest pH value among the four Mollusk specimens, the pH was higher than that of the blank DESS solution, which might have been caused by the dissolution of calcium carbonate from its operculum. Meanwhile, sea cucumbers showed the largest pH decrease despite containing calcareous ossicles, possibly because these skeletal elements are extremely small relative to the overall body size. The impact on the preservative’s pH varied between species and individuals, making it difficult to predict the pH changes. With the exception of the Mollusk specimens, no significant pH decreases were observed, and the pH levels remained above 7.5, suggesting minimal concerns for specimen preservation. However, from a DNA preservation perspective, maintaining a pH of around 8.0 is desirable. For samples where pH fluctuation is a concern, using DESS-NMNS with added Tris-HCl (pH 8.0) would be recommended.

#### 3.1.2. Rate of Evaporation

When accessing historical specimens, we have encountered cases where the complete evaporation of the ethanol preservative rendered the specimens unusable. Although preventive measures such as periodic ethanol replenishment are implemented, preservative solutions with lower volatility and slow evaporation rates are preferable for specimen preservation. Vink et al. (2005), Ferro and Park (2013), and Nakahama et al. (2019) reported that propylene glycol, which exhibits similar dehydrating properties to ethanol, can be used as a low volatility alternative to ethanol for DNA preservation in specimen tissues [37,38,39]. Therefore, we investigated the evaporation volumes of anhydrous ethanol, 75% ethanol, propylene glycol, DESS, and water (as a control) (Figure 3). One milliliter of each solution was placed in 2 mL Eppendorf tubes and left uncapped in the laboratory (RT was approximately 15–21 °C, with 18–23% humidity). Anhydrous ethanol exhibited the highest evaporation rate, with complete evaporation occurring after approximately 3 days (Figure 3a). The evaporation volume of 75% ethanol showed a notable decrease after day 3, with complete evaporation observed after approximately 7 days. This pattern likely reflects the initial evaporation of ethanol, followed by the slower evaporation of the remaining water. Water exhibited an evaporation volume of 60–80 mg/day, while DESS showed a lower rate of 30–40 mg/day, demonstrating approximately half the evaporation rate of water. In contrast, propylene glycol showed an increase in mass, which stabilized after day 5 (Figure 3b). This mass increase can be attributed to the absorption of atmospheric moisture until saturation. Our results indicate that under these conditions, 1 mL of propylene glycol has the capacity to absorb approximately 50–54 mg of water (Figure 3b). While ethanol also possessed dehydrating properties, its high volatility prevented the measurement of its water absorption capacity under these experimental conditions.

To accelerate evaporation and simulate ultra-long-term observation, we conducted evaporation experiments with uncapped 2 mL samples of water, anhydrous ethanol, 75% ethanol, DESS, and propylene glycol at 55 °C (Figure 4). Anhydrous ethanol evaporated completely after approximately 10 h, 75% ethanol after 13.3 h, and water after 37 h (Figure 4a). Under these conditions, 75% ethanol exhibited a linear decrease in mass over time. Propylene glycol also showed a linear mass decrease due to evaporation at 55 °C, with an evaporation rate approximately 15 times slower than that of water. DESS showed a linear mass decrease until 40 h, with an evaporation rate approximately two-thirds that of water, similar to the trend observed at RT, after which the evaporation rate gradually decreased (Figure 4b). These results indicate that the changes in the evaporation rate of DESS were due to the initial evaporation of the water contained in the DESS solution, followed by the slower evaporation of the less volatile DMSO. Thus, in the later stages of evaporation, DESS exhibited behavior similar to that of propylene glycol. Based on these findings, we concluded that even over considerably long periods, the evaporation of propylene glycol during RT storage is negligible, and the DESS specimens are unlikely to completely dry out at RT due to the strong hygroscopic properties of the remaining DMSO. 

After examining the evaporation rate in the uncapped tubes, we next investigated the evaporation rates in the closed storage tubes currently used for specimen preservation at our institution under two conditions (RT and 55 °C) over a two-year period. Propylene glycol was excluded from this investigation because of its negligible evaporation rate. Water and anhydrous ethanol were used as controls, along with DESS. We also compared the tubes with and without Parafilm^®^ (Bemis, Sheboygan Falls, WI, USA) wrapped around their caps as an evaporation prevention measure. At RT, water in the screw-capped tubes (Tokyo Garasu Kikai Co., Ltd., Tokyo, Japan) showed the highest evaporation rate (120 mg over 2 years) (Figure 5, left). In the 1.5 mL tubes, there was no significant difference between the tubes with and without Parafilm, with evaporation amounts of approximately 20 mg over 2 years. DESS showed less evaporation than water, with evaporation amounts of approximately 10–20 mg under all conditions. For ethanol in 1.5 mL tubes, a 10 mg increase in weight was observed over the 2 years. Since the same increase was observed in the Parafilm-wrapped 1.5 mL tubes, this was likely due to moisture absorption by the anhydrous ethanol exceeding ethanol evaporation. In contrast, the samples in the screw-capped tubes showed an approximately 20 mg decrease in weight over 2 years, suggesting that ethanol evaporation exceeded the increase from moisture absorption. Typically, when biological tissues are preserved in ethanol, the ethanol dehydrates the tissues and absorbs water. However, in this experiment, as we tested anhydrous ethanol alone without any specimens or tissues, we could not accurately estimate the evaporation rate of the ethanol preservatives with the specimens. Nevertheless, it is expected that the ethanol in the 1.5 mL tubes would begin to evaporate after reaching moisture saturation.

To evaluate the long-term evaporation rates, each solution was stored at 55 °C to accelerate the evaporation process, as RT evaporation was too minimal for practical assessment. The evaporation rates varied among the solutions, with ethanol showing the highest rate, followed by water and DESS (Figure 5, right). Contrary to expectations, the Parafilm sealing enhanced evaporation at 55 °C, presumably due to the thermal degradation of the Parafilm components at temperatures exceeding its operational range (7–32 °C), compromising the seal integrity. The solutions in the screw-capped tubes exhibited the highest evaporation rates across all solution types (Figure 5, right). After 1 year of incubation, the ethanol had completely evaporated from both the Parafilm-sealed 1.5 mL tubes and the screw-capped tubes, while the unwrapped 1.5 mL tubes showed 78% evaporation. Water retained approximately 50% of its initial volume in the 1.5 mL tubes, while the screw-capped tubes lost approximately 70% of their initial volume. DESS demonstrated superior retention, with the highest evaporation observed in the screw-capped tubes still remaining lower than the other solutions under any condition. After 2 years of incubation, both ethanol and water had almost completely evaporated under all conditions, whereas DESS showed substantial evaporation only in the screw-capped tubes. Notably, while ethanol and water evaporation resulted in empty tubes, DESS evaporation led to the crystallization of its components.

The results indicated that DESS demonstrates excellent pH stability and poses minimal risk of complete sample desiccation. When storing samples in 1.5 mL or 2 mL tubes filled with DESS, simple calculations suggest that it would take at least 90–120 years at RT (10–30 °C) for most of the water content in DESS to evaporate. Given that these tubes are typically made of polypropylene, this timeframe is well within the tubes’ durability limits. While the screw-capped tubes showed higher evaporation rates than the microtubes, and their rubber gaskets have shorter lifespans than polypropylene, microtubes may be more suitable for long-term specimen storage. However, when storing microtubes at ultra-low temperatures such as −80 °C, care must be taken to allow them to warm slightly after removal from the deep freezer before opening, as the tubes can crack if opened immediately after removal from ultra-low temperature storage.

### 3.2. Advantages and Disadvantages of DESS for Morphological Preservation and Taxonomic Studies

#### 3.2.1. Invertebrates


**Nematodes**


The number of known marine nematode species has been reported to be around 10,000 worldwide. Various estimates of undescribed marine nematode species have been proposed based on different evidence, but the small proportion of known species makes it difficult to grasp the complete picture. The most optimistic estimate is approximately 20,000 species [40], while pessimistic estimates, supported by reasonable evidence, range from 1 million to 100 million species [41,42]. Regardless of the actual number, it is undeniable that the proportion of known species is very small compared to the total number of species. Insects are considered to have the highest number of species on earth, with 1 million known species and an estimated total of tens of millions of species when including undescribed ones. The total number of nematode species, including parasitic forms, may exceed that of insects. Moreover, due to the host specificity of parasites, some argue that parasitic nematodes alone could outnumber insect species if we assume that each insect species hosts one unique nematode species [42]. Given such diversity, marine nematodes are expected to possess considerable genetic diversity, making the preservation of specimen DNA crucial for the understanding of biodiversity.

DESS has been used to preserve nematode specimens while maintaining both morphological structure and DNA [17,21]. However, there have been a few actual investigations into the extent of DNA preservation, with only one report demonstrating that DNA remained intact in specimens after 7 months of preservation [17]. We recently published a method for DNA extraction using the supernatant of DESS-preserved nematode specimens and demonstrated that DNA barcoding was possible from the supernatant of 10-year-old preserved specimens [20]. This indicates that trace amounts of DNA leach from the nematode into the DESS solution. To investigate the actual condition of DNA in the specimens, we examined the degree of DNA preservation in the DESS-preserved specimens by extracting DNA from specimens collected between 10 years and 3 months ago (Figure 6). The results showed that DNA fragments larger than 10 kb were preserved in all specimens, with no correlation with the collection year. The highest molecular weight DNA (>60 kb) was found in a sample collected in 2012 (sample 2012-1). In contrast, while the sample from a few months ago showed the largest peak size (28.4 kb), it exhibited overall DNA degradation. These results suggest that DESS prevents significant DNA degradation even after 10 years at RT and that the initial condition of the samples at collection is the most crucial factor for DNA quality. The lack of correlation between DNA quality and the collection year likely reflects the condition of the nematodes at the time of collection or preservation. DNA degrades during the period between collection and DESS preservation, which includes filtering and sorting processes (Appendix A). While factors such as processing time and temperature may have an impact, the DNA within the specimens appears to remain stable once the sample is preserved in DESS. Therefore, the high molecular weight DNA maintained in sample 2012-1 was likely due to factors such as collection in a fresh, viable state, low temperature, and rapid processing before DESS preservation. These results suggest that the reason why the success rate of nondestructive DNA extraction using DESS specimen supernatant [20] is not 100% is that the DNA quality of DESS specimens depends on the condition of the sample when the specimen was created, rather than the collection year. Destructive DNA extraction using part or all of the tissue often yields sufficient DNA for DNA barcoding PCR, even in samples with considerable DNA degradation; therefore, destructive methods remain a practical solution when nondestructive DNA extraction fails.

Based on these results, we confirmed that DESS can preserve not only morphology but also DNA in nematodes. For nondestructive DNA analysis while keeping specimens intact, it is necessary not only to preserve nematodes in DESS but also to collect them in as fresh a condition as possible and to quickly preserve them in DESS. Ogiso-Tanaka et al. (2024) [20] demonstrated that DNA barcoding species identification is possible nondestructively by placing seagrass roots with attached seabed sediment directly into DESS, transporting them to the laboratory at RT, then separating the nematodes, small animals, diatoms, and other meiofauna in the laboratory, preserving each individual in DESS, and using nondestructive DNA extraction from the supernatant.

Immediate preservation in DESS at the field collection site is a good means of preventing DNA degradation in samples. However, caution is required because DESS inhibits DNA degradation through EDTA, a chelating agent that binds divalent metal ions such as calcium (Ca^2+^) and magnesium (Mg^2+^). When sediments containing large amounts of calcium carbonate, which is abundant in the sea, are preserved in DESS, Ca^2+^ binds to EDTA, reducing its effectiveness in inhibiting DNA-degradation enzymes. If samples containing large amounts of calcium carbonate are preserved in DESS and transported from the collection site to the laboratory, it is advisable to either increase the amount of DESS relative to the sample or separate the target organisms promptly (within a few days to weeks) after bringing them to the laboratory without leaving them for too long.


**Arthropods**
-Spiders


Spiders currently comprise 136 families with approximately 53,000 described species [43]. While taxonomic studies utilizing DNA analysis for molecular phylogenetics have become increasingly common [44,45,46,47,48], many historical specimens are unsuitable for genetic analysis. This is because spider specimens were traditionally preserved in 75% ethanol, leading to DNA degradation in older specimens [49]. Currently, preserving specimens in 70–80% ethanol remains standard practice. At these concentrations, COI (cytochrome c oxidase subunit I) analysis and similar studies can often be successfully conducted within approximately 6 months of collection, prompting researchers to either perform DNA analysis as soon as possible or use refrigerated storage to slow DNA degradation. When DNA analysis is planned from the outset, specimens may be preserved in anhydrous ethanol, and since a single leg is sufficient for DNA analysis, a leg may be stored separately after collection. However, specimens preserved in anhydrous ethanol become very brittle due to dehydration, making them prone to damage, such as easily detached legs, so preservation in 70–80% ethanol is preferred for morphological preservation. Therefore, we investigated the use of DESS, which potentially allows for both morphological and DNA preservation, comparing it with ethanol and examining the effects of different concentrations on preserved specimens.

When evaluating specimen firmness on a scale from 1 (softest) to 5 (firmest) (refer to Methods), specimens preserved in anhydrous ethanol became hardened due to dehydration, with all samples showing firmness ratings of 4 or higher (mean 4.5) (Figure 7). During examination, when the legs were grasped with forceps, several samples fractured easily, and the specimen damage from this assessment occurred only in specimens preserved in anhydrous ethanol. Specimens preserved in solutions other than anhydrous ethanol did not break when manipulated with forceps, allowing for leg extension and positional adjustments (Figure 8).

Specimens preserved in 70% ethanol demonstrated elasticity, with firmness ratings ranging from 3 to 3.5 (mean 3.29). Specimens preserved in EDTA-supplemented solutions showed varying degrees of firmness: those with 1 mM EDTA ranged from 1 to 3 (mean 1.85), while those with 10 mM EDTA ranged from 2 to 4.5 (mean 2.95). While EDTA was added to inactivate DNase, it is known to facilitate tissue dissociation by removing divalent ions involved in cellular adhesion [50], potentially contributing to tissue softening. In addition, in this experiment, the variability in tissue firmness may reflect species differences, as various species were used (Appendix A). Specimens preserved in DESS showed firmness ratings from 3 to 4 (mean 3.3), comparable to those preserved in 70% ethanol. Diluting DESS with water tended to decrease specimen firmness: 75% DESS ranged from 2 to 3.5 (mean 2.7), 50% DESS from 2 to 3 (mean 2.5), and 25% DESS from 1.5 to 3 (mean 2.2).

Next, we investigated the DNA quality by extracting DNA from the legs preserved in various solutions. Due to the low DNA concentrations, quality assessment using TapeStation was not feasible. Therefore, we evaluated the extracted DNA by PCR amplification of the COI region used for DNA barcoding, assessing the band intensity of the COI products (Figure 8). The results showed that samples preserved in anhydrous or 75% ethanol exhibited multiple instances of unsuccessful amplification and low PCR product concentrations. In contrast, specimens preserved in 70% ethanol supplemented with EDTA demonstrated PCR amplification yields comparable to those of DESS-preserved specimens over the 6-month period. No substantial differences were observed between undiluted DESS and 75% DESS (Figure 8). These results indicated that specimens preserved in anhydrous ethanol underwent DNA degradation due to diminished dehydration effects caused by the spider body fluids, as the ethanol was not replaced during storage. Similarly, samples stored in 75% ethanol likely experienced DNA degradation during RT storage. Conversely, the addition of EDTA to 70% ethanol inhibited DNA degradation despite RT storage. While the current results showed no discernible differences in PCR amplification between the 1 mM and 10 mM EDTA concentrations, variations may become apparent over extended storage periods.

Although 70% ethanol is commonly used for specimen and tissue transportation due to compliance with airline and shipping regulations, concerns about its adverse effects on DNA have been raised. The finding that EDTA supplementation inhibits DNA degradation suggests its utility for future transportation purposes. In addition, 70% ethanol remains unfrozen at −80 °C, allowing for post-transport frozen storage. Regarding the EDTA concentration in 70% ethanol, a 10 mM concentration is recommended for RT storage as a precautionary measure, given its potential for enhanced DNA preservation.

Further testing is required to determine the optimal conditions for the long-term RT storage of spider specimens that maintain both morphological and DNA integrity with minimal maintenance. We look forward to the extended observations and trials by future arachnologists.

Next, we evaluated the potential for nondestructive DNA extraction using the DESS supernatant [20]. DNA extraction was performed on specimens preserved in 25–100% DESS solutions (as shown in Figure 9), with COI amplification successful in 68.8% of samples (66/96 samples). Sequence verification confirmed that 100% of the successfully amplified samples contained authentic spider COI sequences. While our 1-month preservation study revealed no clear correlation between PCR success rates and either DESS concentration or spider species, extended preservation periods might yield improved success rates. As DNA extraction from the DESS supernatant represents a completely nondestructive method that eliminates the need for proteinase treatment, this approach warrants consideration, particularly for valuable specimens.

-Insects

Insect specimens are typically preserved in a dry state [51]. However, since dried specimens are not suitable for DNA preservation, it is common practice to preserve a single leg in ethanol for DNA analysis, especially for larger insect species. Due to ethanol’s high volatility, propylene glycol has been proposed as an alternative preservative due to its low evaporation rate [37,38,39]. Like ethanol, propylene glycol protects DNA through the prevention of oxidation by limiting air exposure and the inhibition of DNase activity through dehydration.

In this study, legs from the same individual of *Prionus insularis* (a species of long-horn beetle) were divided into 12 parts (femora and tibiae) and preserved under various conditions for 5.5 months to evaluate the DNA preservation efficacy (Figure 10). As a result, DNA extraction from the tibiae yielded poor results, with DNA concentrations falling below the TapeStation detection limit, making it impossible to evaluate the differences in DNA quality across varying ethanol concentrations. In contrast, among DESS-preserved tibiae, while the DNA was below the detection limit in 25% DESS, DNA bands were detected in the 50% DESS-preserved specimens, albeit at concentrations too low for DIN value calculation. Tibiae preserved in 90% DESS showed relatively high molecular weight DNA with a DIN value of 5.8, although below the recommended concentration for DIN calculation. In contrast, the femora yielded sufficient DNA for analysis. Condition 7, where the specimen was dehydrated with ethanol and then air-dried, showed relatively high molecular weight DNA with a DIN value of 5.8. In condition 8, where the specimen was dried after immersion in a mixture of 70% ethanol and DESS (condition 3), the specimen showed DNA degradation to 200 bp–1 kb fragments with a DIN value of 1.6. Similarly, in condition 12, where the specimen was dried after immersion in 90% DESS (condition 6), the specimen showed DNA degradation to 400 bp–2 kb fragments with a DIN value of 2.2. These results indicate that DNA degradation progresses in the presence of moisture, even with EDTA present, when specimens are not fully immersed in the liquid. This finding emphasizes the importance of complete dehydration when preparing dried specimens. It should be noted that because all specimens were stored in dark conditions, the effects of light exposure could not be evaluated. Furthermore, specimens preserved in DESS showed the most effective prevention of DNA degradation, with even 5% DESS showing better results than dried specimens. Although the effects of different preservation conditions on DNA integrity were consistent across independent specimens, it was difficult to prepare multiple samples under identical conditions for comparison. Therefore, future practical research will need to determine what concentration of DESS can achieve effects comparable to 100% DESS. While 100% DESS is currently the optimal preservation method, achieving comparable results with diluted solutions would make the method more cost-effective and practical.

Next, we investigated what happens when insects are placed in DESS. To exclude potential ethanol effects, the insects were immediately immersed in DESS after collection. We observed that the insects remained alive for some time and continued to move around in the DESS solution. This observation suggested that for insects, it is better to kill them with ethyl acetate before placing them in DESS rather than immersing them alive. Specimens killed using ethyl acetate vapor show more extensive DNA degradation compared to those euthanized with ethanol [52], although ethyl acetate does not directly cause DNA damage. The enhanced DNA degradation in the ethyl acetate-treated specimens can be attributed to the solvent’s insufficient dehydrating properties compared to ethanol, resulting in inadequate suppression of the DNA-degrading enzyme activity. After being placed in DESS, the specimens initially floated, presumably due to the high specific gravity of the DESS (Figure 11). However, after some time, possibly due to the complete penetration of DESS into the body, the specimens gradually sank (Figure 12a). In addition, since it is impractical to preserve each individual in separate tubes during collection, we evaluated the feasibility of batch preservation by placing multiple specimens into a single 25 mL tube containing DESS immediately after collection (Figure 11 and Figure 12a). After 5 days of preservation, the 15 mL DESS solution containing five insects exhibited brown discoloration, with all specimens floating on the surface. When individually transferred to 1 mL of 90% DESS, the specimens persistently remained at the surface, even after gentle mixing by inverting the tubes (Figure 11).

Next, we investigated the effectiveness of nondestructive DNA extraction [20] from DESS supernatant in insects. First, the specimens were placed in 100% DESS immediately after collection and stored at RT. After 5 days, DNA was extracted from the supernatant, and PCR amplification of the COI was attempted, but no amplification was observed. Therefore, we reduced the DESS concentration to 90% and continued the storage at RT. We then used specimens that had been preserved together in the same DESS solution for 5 days before being separated into new individual 2 mL tubes containing fresh 90% DESS (Figure 11). After 1 day of individual storage in 90% DESS, we again attempted nondestructive DNA extraction from the supernatant, followed by COI PCR amplification, but no amplification was observed. In addition, when we tested the supernatant DNA extraction and COI PCR amplification of large beetles (2–3 cm in length) after several days of DESS preservation, no COI amplification was detected. These results demonstrated that DNA extraction from the DESS supernatant after several days of preservation was insufficient to obtain adequate template DNA for PCR amplification in insects. This finding simultaneously suggested that when multiple specimens are initially stored together in the same DESS solution during collection, the risk of DNA cross-contamination between individuals is likely to be minimal.

Subsequently, we reduced the DESS concentration to 90% and stored specimens at RT in dark conditions for 2 years, then performed DNA extraction and PCR using the same method (Figure 12). Using the LCO1490/HCO2198 primer pair, 35% of samples (7/20 samples) showed clear amplification bands, and two additional samples showed weak bands that yielded COI products after purification and use as PCR templates, resulting in a total success rate of 45% (9/20 samples). To evaluate the PCR amplification success with shorter COI fragments, we also tested the mlCOIint/HCO2198 primer pair, which resulted in clear amplification bands in 55% of samples (11/20 samples) and a weak band in one additional sample, achieving a total success rate of 60% (12/20 samples) for the COI amplification products. The lack of correlation between PCR success and insect body size suggested that the specimen’s condition at the time of collection and morphological differences between individuals might influence the results (Figure 12). These results suggest that collecting multiple insects in a single tube/bottle containing 100% DESS and later individual separation in the laboratory does not pose contamination issues at the DNA barcoding level. This method allows for the preservation of fresh specimens by quickly storing them in DESS during collection. Furthermore, nondestructive DNA extraction using DESS supernatant is a particularly convenient method for small insects, where separating legs for DNA extraction is challenging. Currently, with success rates ranging from 35–60%, destructive methods remain the most reliable way to obtain DNA. However, for specimens that need to be preserved intact, this nondestructive method is worth trying as a first option. In addition, empirical evidence suggests that diluting the DESS concentration with water increases the success rate, which warrants further investigation in future studies.

Next, we investigated the effects of DESS on specimen coloration. In ethanol specimens, it is known that the green coloration on the insect body surface fades [53,54,55], and as shown in Figure 13a,b, grasshoppers had already experienced significant fading after being placed in ethanol for 2 months (Figure 13a). In DESS or DESS-NMNS, the solution became an ochre color with brown tones, which was different from the yellowish-green color observed in ethanol. There was minimal difference in body coloration between DESS and DESS-NMNS, indicating that antioxidants and Tris-HCl had no effect on grasshopper coloration. Based on these properties, DESS cannot be expected to preserve dobby coloration in species containing pigments across other taxonomic groups. However, it can be used without issue for insects with structural coloration.

As mentioned previously, insect specimens are typically stored as dried specimens in museums and laboratories [51]. Since DESS contains not only salts that cause iron rust but also EDTA, a chelating agent that dissolves iron at high concentrations, creating dried specimens with these residual components not only interferes with morphological observation but also causes specimen pins to rust [56]. Therefore, when creating dried specimens from DESS-preserved specimens, desalination is necessary. To verify this method, we used the excised wings of *Protaetia orientalis*, which have characteristic structures on their elytra, and performed scanning electron microscopy (SEM) after desalination with water and 70% ethanol. We also investigated the effect of brushing versus no brushing to remove surface contaminants. As a result, despite washing only once with water and 70% ethanol at 100 times the weight of the wing (approximately 0.02 g), no salt crystals were observed under the SEM (Figure 14). When desalinating the entire body, including cavities, it is expected that much larger amounts of water or 70% ethanol would be needed, but for observing only the wings, a single washing might be sufficient. Since the wings floated or repelled water during water washing, washing with 70% ethanol was easier than with water. Therefore, it seems best to perform washing with 70% ethanol (or possibly even lower concentrations of ethanol). Furthermore, there was a significant difference in the amount of residual contamination between cases with and without brushing (Figure 14, red arrows). Therefore, when conducting morphological observations under an electron microscope, brushing of the insect surface is necessary. Since no crystal structures derived from salt were observed under SEM, these contaminants are considered to be derived not from DESS but from the organism itself or its living environment.


**Shrimp (*Neomysis*)**


*Neomysis japonica* belongs to the order Mysida (Arthoropoda, Crustacea) and inhabits brackish and freshwater areas throughout Japan. Globally, it is also distributed in Siberia and along the Pacific coast of North America. While formalin is commonly used for the morphological preservation of mysids [57], it has the disadvantage of making DNA extraction difficult. In addition, ethanol, which is widely used for arthropods, is not frequently employed for *N. japonica* as it turns the specimens white (Figure 15). When the specimens were preserved in DESS solutions and morphological differences were observed, it was found that the body color and appendages were maintained (Figure 15). Because DESS does not cause hardening through protein coagulation or cross-linking like ethanol or formalin does, *N. japonica* specimens could be maintained in a soft state.

Next, we examined whether DNA could be extracted from the supernatant of the DESS solution. Based on preliminary experiments with insects, which showed higher DNA extraction success rates from specimens preserved in diluted DESS compared to 100% DESS, we reduced the DESS concentration for *N. japonica* specimens from 100% to 50%. DNA was then extracted using 700 µL of the supernatant after 18 days of preservation (Figure 16a). While PCR amplification using Folmer’s COI primer (LCO1490/HCO2198 [30]) did not yield any PCR products across all samples, amplification was successful in all samples when shorter COI primers (mlCOIint/HCO2198 [31]) were used (Figure 16b). Sequence analysis of the PCR products confirmed that *N. japonica* DNA was obtained. To investigate the effects of longer preservation periods, 100% DESS was added to achieve a final concentration of approximately 90%, and the specimens were preserved at RT for an additional 2 years. During this period, no significant morphological changes were observed. When DNA was re-extracted from the supernatant of specimens preserved in 90% DESS for 2 years, the COI was successfully amplified in all samples using both primer pairs (Figure 16c). These results suggest that DNA elution from the specimens into the DESS solution initially occurs with shorter fragments, and longer preservation times in DESS allow for the accumulation of longer DNA fragments in the solution. Even if PCR amplification from the specimen supernatant with reduced DESS concentration is unsuccessful, extended preservation in DESS may increase the likelihood of obtaining DNA. Therefore, nondestructive DNA extraction using the DESS supernatant could be an effective method for valuable specimens that should not be damaged or for small-sized samples. The higher success rate compared with insects might be attributed to the very low water content in insects. This method is expected to be particularly suitable for nondestructive DNA extraction from aquatic organisms, and further validation studies are anticipated.


**Sea cucumbers**


Sea cucumbers encompass approximately 1800 named species within the class Holothuroidea (phylum, Echinodermata) [58], and holothurians are the dominant animals in deep-sea benthic communities (e.g., [59]). In general, sea cucumber specimens for the morphological identification of taxonomic and biodiversity studies are fixed with neutralized 3.7% formaldehyde and preserved in 70% ethanol (e.g., [60]) or directly preserved in 70–99% ethanol (e.g., [61]). For molecular studies, small pieces of tissue (1–2 mm^3^) cut from the tentacles, tube feet, muscles, or even gonads is preserved in absolute ethanol (99.5%) and stored in a refrigerator (ideally at −80 °C) to avoid any degradation of the DNA (e.g., [60]).

Although DESS containing EDTA is expected to dissolve calcareous ossicles within the body wall due to its chelating effect, the body surface of the sea cucumber is covered by a thick body wall composed mainly of collagen and a cuticular layer. Therefore, it is of interest to investigate how DESS affects these tissues and whether it is effective for DNA preservation. To address these questions, three species of sea cucumber (*Elpidia kurilensis*, *Peniagone* sp., and *Pseudostichopus* sp.) were immediately preserved after sampling either in 99% ethanol and frozen for 1 month or in 100% DESS at RT for 28 months. Morphological observations of the specimens and their calcareous ossicles revealed that the ossicles, which were observable in the ethanol-preserved specimens (Figure 17a,d,g), had dissolved in DESS (Figure 17j,p). In *Elpidia kurilensis*, which possesses robust ossicles, some morphological features of the ossicles remained preserved in DESS (Figure 17m). However, in *Peniagone* sp. and *Pseudostichopus* sp., the ossicle morphology was severely degraded, with little resemblance to their original structure.

Regarding overall body morphology, ethanol-preserved specimens maintained their shape well with fixed tissues, whereas DESS-preserved specimens showed transparency of the body surface with visible intestinal walls, and the entire tissue had softened to the point where it would disintegrate upon touch (Figure 17k,l,n,o,q,r). These results clearly demonstrate that DESS is unsuitable as a preservation solution for maintaining the morphology of sea cucumbers. Furthermore, these findings indicate that preservation solutions containing EDTA as a DNA degradation inhibitor cannot be effectively used for maintaining the morphology of sea cucumbers.

Next, we examined the effectiveness of the tissue preservation solutions for the DNA study. Sea cucumbers contain high water content and are prone to autolysis by endogenous proteases and DNases after death, making it desirable to preserve specimens in preservation solution immediately after sampling. Mucharin et al. investigated the DNA preservation effects in sea cucumber body wall tissues preserved in ethanol and RNAlater (currently RNAprotect Tissue Reagent, QIAGEN) under refrigeration, and silica gel-dried tissues at RT in tropical climates where refrigeration is challenging [54]. They demonstrated that in the short term, drying can preserve sufficient DNA for DNA barcoding at a low cost [62]. However, unlike plant specimens in which silica gel drying is commonly used, sea cucumbers have a significantly higher water content. Consequently, achieving sufficient dryness with silica gel to suppress DNA degradation is only feasible when dealing with a limited number of sampled specimens. In our study, we extracted DNA from the tissue sections of six sea cucumber species preserved in DESS for 2 years under dark conditions (Figure 18a). The DESS-preserved tissues had become very soft, allowing tissue samples to be obtained simply by grasping with forceps. High molecular weight DNA was preserved in both the transparent tube foot sections and the tissue containing the gut walls or contents (Figure 18b). However, in sample 13 (*Pseudostichopus* sp.), DNA degradation was relatively advanced, with DIN values of 3.9 for intestinal tissue-derived DNA and 5.5 for tube foot-derived DNA. As sample 13 was the largest specimen among the tested samples, we suspect this was due to the high tissue-to-DESS solution ratio. In addition, as shown in Figure 17, DESS dissolves the taxonomically important morphology in sea cucumbers, calcareous ossicles. Initially, we thought the poor DNA quality of sample 13 was due to the reduced chelating capacity caused by dissolved ossicles. However, because samples 12 and 13 only possess ossicles around the oral tentacles, this possibility is low. It is more likely that the specimen happened to be in poor condition before preservation, or simply that its large body size contained more water content, or that DESS penetration took longer due to the specimen’s size. The results demonstrate that DESS is an effective preservation solution for storing tissue sections as DNA samples from sea cucumbers. To maintain high-quality DNA, two key points should be considered: (1) ensure that the tissue volume is not excessive relative to the DESS volume, and (2) avoid including ossicles in the preserved tissue. By following these precautions, the DNA degradation inhibitory effect of DESS can be maximized.

**Slugs and snails** (Gastropoda)

Gastropod specimens for molecular analyses are typically preserved in 70–100% ethanol, often with shells. Before fixation, it would be better to divide the soft bodies from the intact shells via the niku-nuki method [63]. In this method, live individuals are boiled in water for a few minutes, and the muscles are separated from the shells using forceps. This study showed that this method is effective for DNA sequencing as well as morphological analyses because it can deactivate DNase and molluskan mucus containing mucopolysaccharides, which obstruct DNA amplification. The remaining shells are sometimes washed and dried.

We investigated the effects of DESS preservation compared with ethanol on shell morphology and DNA in the soft bodies of gastropods. The collected specimens of four species, *Lottia kogamogai*, *Echinolittorina radiata*, *Nassarius siquijorensis*, and *Sinotaia quadrata histrica*, were preserved in DESS solution or 100% ethanol (see Section 2.1.1). The conditions of the morphology and DNA in DESS were compared with those in ethanol.

The quality of the extracted DNA from each specimen is shown in Figure 19. DNA from *L. kogamogai* and *E. radiata*, which were immersed in DESS for 3 months with their shells, was found to be quite fragmented, whereas long DNA chains remained in tissues of *N. siquijorensis* preserved for 2 months in 100% ethanol. Calcium ions would be dissolved from the calcareous shells by DESS’s chelating effects, which were considered to lead to the activation of DNA fragmentation. To verify the effects of DESS on molecular samples with shells, the degrees of fragmentation were compared among the four types of preservation, DESS or ethanol, and the presence or absence of intact shells, using *S. quadrata histrica* (Figure 20c,d). Although the results were less obvious than those of the previous experiment, DNA extracted from the specimen with a shell preserved in DESS was more fragmented than the others. The DESS-preserved soft body without the shell was in as good condition as the ethanol-preserved specimens. These results suggest that DESS preservation for molecular analyses is sufficiently useful only when soft bodies are separated from intact shells by the niku-nuki method and immersed without their shells.

Chelating effects had an extensive influence on the gastropod shell. The shell of *E. radiata* immersed for 4 months in DESS had almost completely disintegrated (Figure 19). Although the shell of *L. kogamogai*, which is relatively thinner than that of *E. radiata*, was less dissolved and became tender and broken. Their soft bodies remained more tender than those in ethanol preservation, but were not broken. The DESS solution also dissolved a shell of *S. quadrata histrica* immersed for a month, whose thick spire was eroded and had large holes, whereas the ethanol-preserved one was not dissolved. A DESS-preserved soft body was not so collapsed and kept the conditions close to the fresh state without dehydration, but was more tender and fragile than the specimen preserved in ethanol.

These results showed that the DESS solution has extensive negative influences on the molecular and morphological analyses of gastropods if the shells are immersed together. It would be fine for at least molecular and morphological analyses to preserve only the soft body, although it remains fragile. If the DESS solution is used for the preservation of the gastropod, only the soft body must be immersed after separation from the shell, and be treated gently so as not to break. Thus, we cannot help but say that this solution is not less suitable for field research of gastropods, unless niku-nuki processing is carried out immediately after specimen collection.

#### 3.2.2. Vertebrates


**Birds**


At the National Museum of Nature and Science, Japan (NMNS), as with other species, bird blood or muscle tissue samples are preserved in ethanol and stored frozen at −80 °C as DNA tissue specimens. A recent study by Lecce et al. [64] reported that ethanol is the most commonly used preservative for bird blood samples, likely due to its low cost and nontoxic nature. Several preservation solutions have been developed specifically for DNA preservation. For blood preservation, these include Queen’s buffer (pH 7.5), consisting of 1% N-lauroylsarcosine, 0.01 M EDTA, 0.01 M NaCl, and 0.01 M Tris [15], and Longmire’s buffer, composed of 0.5% SDS, 10 mM NaCl, 100 mM EDTA (pH 8.0), and 10 mM Tris-HCl (pH 8.0) [65]. For tissue preservation, Seutin et al. [15] indicated that DESS is the most reliable preservative. They demonstrated that blood samples preserved in Queen’s buffer or tissue samples preserved in DESS maintained DNA quality comparable to samples stored at −70 °C, even after 24 weeks at RT, recommending these methods for field collection and facilities without deep freezers. The choice between ethanol and Queen’s buffer or DESS for blood and tissue preservation often depends on the researcher’s preference. In our laboratory, when using DESS, samples are transported at RT, stored refrigerated until DNA extraction, and any remaining blood samples are frozen. When using ethanol, samples are transported at RT, stored refrigerated or frozen until DNA extraction, and frozen for long-term storage.

Since the DNA preservation efficacy of DESS has already been verified for bird blood and other animal tissues [15], we investigated the tissue penetration properties of DESS and other preservative solutions, assuming the preservation of the DNA of the bird muscle tissue. For long-term DNA preservation within tissues, the preservative solution must effectively penetrate the tissue core. For instance, it is known that in animal specimen preparation using formalin, inadequate fixation of the tissue interior can lead to progressive decay. Therefore, we conducted a comparative study using anhydrous ethanol, 10% formalin, propylene glycol, and DESS, with chicken breast muscle as the sample tissue.

Pieces of muscle tissue were placed in 50 mL of each preservative solution containing red ink at RT (approximately 15 °C), and the changes were monitored. When tissues were placed in the solutions, they immediately sank to the bottom of the beaker, except in the DESS solution, where the muscle tissue remained afloat. This difference was attributed to the varying specific gravity of the solutions. After 24 h, the muscle tissue fragments in ethanol had become firm and showed shrinkage and hardening due to dehydration (Figure 21). Cross-sectional examination revealed that the surface tissue was most severely hardened, and while the tissue surface was stained with red pigment, only a few millimeters from the surface showed white dehydration, with no visible changes in the central portion. This suggested the possibility of DNA degradation in the interior of the tissue.

In the muscle tissue preserved in 10% neutral buffered formalin, moderate tissue firmness was observed, although not to the extent of the hardening seen in the ethanol-preserved tissues. Cross-sectional analysis revealed pigment penetration to approximately 5 mm depth from the surface, while the central region showed no visible color changes (Figure 21). The absence of gas bubble formation and effective decay suppression suggested substantial penetration of the formalin fixative into the tissue interior.

Muscle tissue preserved in propylene glycol showed numerous gas bubbles, indicating ongoing decay in the solution. The muscle tissue was slightly hardened but remained softer than the formalin-preserved tissue. Cross-sectional examination revealed red staining throughout the interior, demonstrating that propylene glycol had penetrated the entire tissue (Figure 21). However, it failed to immediately halt the decay process. When the tissue was returned to propylene glycol after photography on day 1, no further bubble formation was observed, suggesting some ability to arrest decay once it had begun. While propylene glycol is known for its preservative properties and is used as an additive in food and cosmetics, it apparently lacks rapid sterilization capability, possibly due to the large tissue volume in this case. The absence of decay in the ethanol-preserved tissue, which also has a dehydrating effect, can be attributed to ethanol’s stronger antimicrobial properties. Although the optimal concentration for ethanol’s antimicrobial effect is reported to be around 70% [66], and higher ethanol concentrations are said to reduce antimicrobial activity, recent detailed studies have shown that anhydrous ethanol maintains high antimicrobial efficacy when used with water-containing samples [67]. In our experiment, rather than temporary ethanol sterilization, we stored water-containing muscle tissue in anhydrous ethanol, which likely provided sufficient antimicrobial effect to completely inhibit decay.

In the DESS-preserved muscle tissue, the tissue fragments initially floated in the solution for the first few hours. While the tissue was still floating at the 6 h observation point, it had sunk to the bottom of the DESS solution after overnight storage. This was likely due to the complete penetration of DESS into the tissue fragments overnight, which was further supported by the observation that the ink had penetrated throughout the cross-section, resulting in uniform red coloration (Figure 21). In DESS preservation, the high cell permeability of DMSO appeared to facilitate the distribution of EDTA and NaCl throughout the tissue interior, where the high salt concentration suppresses tissue decay while EDTA, a DNA degradation enzyme inhibitor, protects the DNA.

These results indicate that when using ethanol as a preservative solution for DNA tissue sections, it is essential to limit the tissue volume to allow sufficient dehydration by ethanol and achieve rapid tissue dehydration through techniques such as preparing thin tissue sections. Similarly, for DESS preservation, thinner tissue sections are preferable to bulk tissue to facilitate the rapid cellular penetration of the preservative.

In our experimental trial with propylene glycol, while the initial tissue decay might be attributed to excessive tissue volume, our dehydration experiments (Section 3.1.2) estimated that propylene glycol has a dehydration capacity of 50–54 mg of water/mL. Therefore, the 50 mL of propylene glycol used in this experiment had a theoretical dehydration capacity of 2.5–2.7 g, suggesting that the tissue volume was not necessarily excessive relative to the preservative volume. This indicates that, considering the rate of cellular penetration, preparing thin tissue sections rather than bulk tissue is crucial for ensuring effective dehydration, regardless of the total tissue mass.

Propylene glycol, like DMSO in DESS, is known for its high cellular permeability, although studies using human tissue have shown a slightly slower cellular penetration rate at lower temperatures compared to DMSO [68]. However, in our experiment conducted at RT (approximately 15 °C), the penetration rates were likely comparable. Propylene glycol, which is also used as a safe food additive, has been proposed as an alternative preservation solution to ethanol because of its minimal evaporation and preservative properties. Nakahama et al. reported a method of preserving insect legs in propylene glycol after dehydrating them with ethanol for DNA analysis [39]. However, propylene glycol itself does not have a direct DNA protective effect; it only provides an indirect effect through the inhibition of DNA-degrading enzymes by dehydration. Therefore, it may not be optimal for the long-term preservation of muscle tissue with a high water content. While DMSO is known to protect DNA from photodamage [69], there is currently no evidence that propylene glycol possesses such functionality. Therefore, when using propylene glycol, greater care must be taken to avoid exposing the tissue specimens to light. Studies have reported that the optimal DNA protective effect of DMSO is observed at a concentration of approximately 2% [69]. Therefore, creating a modified DESS solution containing 2% DMSO with propylene glycol might achieve both cellular permeability and DNA protection while minimizing toxicity. From a cost perspective, there is no significant difference between propylene glycol and DMSO. Using nontoxic solvents such as propylene glycol instead of DMSO, which exhibits low cellular toxicity at RT, may enable safer utilization of preservation solutions. Further verification of this possibility is needed.

Next, we investigated the rate of DNA degradation in tissue samples stored at natural temperature in various preservation solutions, simulating the conditions during tissue transportation. When transporting ethanol-preserved specimens domestically or internationally, ethanol concentrations must comply with regulatory limits. In Japan, many shipping companies permit the transportation of specimens in up to 70% ethanol, making this concentration common for tissue or specimen preservation during shipping. While it is known that DNA in specimens preserved in 70% ethanol becomes difficult to PCR-amplify after several months at RT, the precise rate of DNA degradation has not been thoroughly investigated. Therefore, using chicken breast muscle as a model, we examined the extent of DNA degradation in samples preserved in 70% ethanol over typical transportation periods.

Portions of chicken breast muscle (wet weight approximately 0.1–0.2 g) were placed in seven different preservation solutions and stored at 25 °C. DNA was extracted from tissue pieces (approximately 1.5 mm^3^) daily from day 0 to day 4 and again on day 10 to assess the DNA quality (Figure 22). After 24 h, the tissue pieces in ethanol and propylene glycol had become very hard due to dehydration. Tissue pieces in 70% ethanol and DESS remained soft even on day 10 of preservation (Figure 22a). After 10 days, white precipitates were observed in the muscle tissue stored in 70% ethanol containing EDTA and in propylene glycol (Figure 22a). When the tubes were inverted for mixing, fine tissue fragments dispersed into the solution from the muscle tissue stored in 70% ethanol containing EDTA (Figure 22b). This was likely because the cell membranes were destroyed by 70% ethanol, and additionally, the cell-to-cell adhesion was weakened by the effect of EDTA [50]. Muscle tissue in propylene glycol also showed white material spreading in the solution, although the nature of this material remains unknown. For DNA extraction, precipitated tissue fragments were not used; rather, tissue pieces were retrieved with forceps, and approximately 2 mm sections were cut out with scissors for DNA extraction. The results showed little difference in the degree of DNA degradation in tissues preserved in ethanol and 70% ethanol up to day 4 at 25 °C (Figure 22c). Similar low levels of DNA degradation were observed in other preservation solutions. These results indicate that DNA degradation is substantially inhibited within 4 days when transporting samples in 70% ethanol at the volume ratio used in this study (1.8 mL of 70% ethanol: 0.2 g of tissue). When tissue sections were excised for DNA extraction on day 10, significant DNA fragmentation was observed in tissues preserved in 70% ethanol. In contrast, DNA degradation was inhibited in tissues preserved in 70% ethanol supplemented with either 1 mM or 10 mM EDTA. These results demonstrate that adding EDTA at concentrations of 1 mM or higher enables the safe transportation of tissues for DNA analysis while complying with ethanol concentration regulations. In addition, because 70% ethanol does not freeze at −80 °C, samples can be stored frozen directly after transportation at facilities equipped with deep freezers. Generally, samples transported in 70% ethanol are transferred to anhydrous ethanol before freezing, but this extra step can be eliminated. Although 1 mM EDTA showed sufficient effect in this short-term experiment, over longer periods, as ethanol concentration decreases due to evaporation, DNA degradation enzymes may become activated in samples not kept frozen or refrigerated. Therefore, an EDTA concentration of 10 mM is presumed to be safer. This solution can be easily prepared by adding 4 mL of commercially available 500 mM EDTA to 100 mL of 70% ethanol. In addition, based on the ethanol evaporation experiment results, when storing ethanol specimens at RT, we recommended using 1.5 mL or 2.0 mL tubes typically used in molecular biology experiments rather than screw cap products with rubber gaskets that allow faster ethanol evaporation.

Differences in the tissue dissolution rates were observed during the process of dissolving the tissue pieces in lysis buffer for DNA extraction. After 5 h of incubation with shaking in a 55 °C water bath, tissue pieces from most preservation conditions had completely dissolved in lysis buffer containing proteinase K. However, tissue pieces preserved in propylene glycol were not completely dissolved and required overnight incubation for complete dissolution. This indicated that the dissolution time of the tissue may vary depending on the preservation conditions.

Next, we investigated what happens to the DNA in tissues when the preservative solution evaporates. We forcibly evaporated ethanol or DESS in 2 mL tubes containing bird muscle tissue under negative pressure to create a condition with almost no solution. In the ethanol samples, a small amount of water originally contained in the muscle remained, while in the DESS solution, a large amount of salt precipitated (Figure 23). After keeping these samples at 25 °C for 10 days, we cut out part of the tissues for DNA extraction. We found that the DNA in the tissues that had been in ethanol was degraded, whereas the evaporation of the preservative solution had almost no effect on the DNA quality in the DESS-preserved samples. These results indicated that DESS prevents DNA degradation caused by preservative evaporation during long-term storage at RT and can preserve DNA even if replenishment or replacement of the preservative solution cannot be performed when evaporation occurs. While evaporation of the preservative solution increases the risk of oxidation when the tissue is exposed to air, DESS contains DMSO, which has strong hygroscopic properties, preventing complete drying and minimizing this risk.

We investigated what changes occur when tissue sections are preserved at RT in DESS for an extended period. While it has been demonstrated that specimens immersed in DESS solution maintain a soft state for other organisms (spiders, insects, mysids, and sea cucumbers), it was unclear what changes would occur in tissue sections such as muscle. Therefore, we observed bird muscle sections preserved in 70% DESS at RT in the dark for 2 years (Figure 24). Although there were slight differences depending on the tissue, cell fragments that appeared to have detached from the surface were observed at the bottom of the tube in all tissues. This was predicted to be the result of cell dissociation occurring due to the chelating action of EDTA inhibiting the function of cell adhesion proteins [50]. When we attempted to extract DNA from these suspended tissue fragments, we were able to obtain DNA of similar quality to that extracted from the cut muscle tissue (700 ng–30 μg), albeit in smaller quantities (Figure 24a–c). The DNA obtained from the suspended cells had a lower concentration, possibly due to the smaller number of cells, but with Qubit measurement values of approximately 5–15 ng/μL (100–300 ng), it was sufficient for use as a PCR template for applications such as DNA barcoding. The precipitated suspended tissue can be collected by taking the suspended tissue along with DESS using a pipette and centrifuging to collect the cells (Figure 24b), which reduces the risk of DNA contamination as it eliminates the need for scissors or scalpels. Meanwhile, when DNA was extracted from tissue fragments that had adhered to the lid and were not immersed in the DESS solution (Figure 24a, SM-3), the DNA maintained its quality. This result indicates that once DESS penetrates the tissue fragments, DNA degradation is inhibited even when the fragments are not immersed in DESS. While it is optimal for tissues to be completely immersed in the preservative solution, there may be cases where tissues are not immersed in the preservative solution due to vibrations or movement during transportation, but the impact on DNA is likely minimal. In addition, when there are limitations on the amount of preservative solution that can be transported, it may be possible to minimize the amount of DESS relative to tissue. Furthermore, 70% DESS not only can be stored at RT and does not freeze at −80 °C, making it suitable for storage in deep freezers alongside other ethanol-preserved liquid specimens.

#### 3.2.3. Seagrasses and Algae


**Seagrasses**


The methods of collecting samples for DNA and extracting DNA from aquatic vascular plants, including seagrasses, are not fundamentally different from those for terrestrial plants. However, submerged leaves lose moisture from their plant bodies more quickly than terrestrial species because of the absence of a cuticle on the leaf surface. When transporting the collected samples, it is necessary to seal them in plastic bags to prevent temperature increases. Additionally, because filamentous algae or calcium deposits are often found on the leaf surfaces of submerged leaves, selecting leaves free of such attachments or washing the samples with freshwater after collection to remove them is required.

Inglis and Waycott [70] stated that the most common problem for DNA extraction is that drying does not occur rapidly, resulting in degraded DNA. They suggested drying the tissue samples in vials filled with silica gel. Although it is essential to use silica gel for rapid drying, the leaves of some seagrass species, such as *Halophila* and *Halodule*, tend to break into fragments in silica gel. Therefore, it is recommended to wrap the leaves in a nonwoven fabric bag and seal them together with silica gel in tubes or zip-lock bags.

When *Zostera marina* and *Phyllospadix iwatensis* were stored in 100% DESS, pigments leached out and the DESS turned brown, with this effect being particularly pronounced in *Z. marina* (Figure 25a–c). After 7 months of 4 °C or RT storage in DESS (Figure 25d), DNA extraction revealed that both species maintained relatively long DNA fragments with a DIN value of 5.7 and molecular weight peaks at approximately 9 kb (Figure 25e). In *P. iwatensis*, no difference in DNA degradation was observed between samples stored at 4 °C and RT for 7 months (Figure 25f). The samples were then stored at RT for an additional 8 months in either 50% or 100% DESS before DNA extraction. For comparison, we used DNA extracted using our standard method: drying with silica gel, followed by DNA extraction. For *Z. marina*, DNA extracted from the leaves after 1 month of drying showed significant degradation with a DIN value of 1.0 (Figure 25g). In contrast, samples preserved in DESS for 15 months (including 8 months at RT) showed similar DIN values ranging from 2.9 to 4.9 in both 50% and 100% DESS concentrations. These results suggest that the leaf position and maturity used for DNA extraction had a greater impact than the DESS concentration; however, DNA of usable quality was maintained even in 50% DESS at RT For *P. iwatensis*, DNA extracted from samples dried for 11 days and 1 month showed DIN values ranging from 5.9 to 6.4 and 1.0 to 2.8, respectively, indicating high-quality DNA preservation in samples dried for 11 days. In comparison, samples preserved in DESS for 15 months at RT showed no difference between 50% and 100% DESS concentrations, with DIN values of 2.9 to 4.9, respectively. These results indicate that while DESS does not preserve DNA in aquatic plants as effectively as it does in animal cells, it provides a more effective preservation method than drying when immediate DNA extraction is not possible. Furthermore, although no difference in DNA quality was observed between samples stored at 4 °C and RT during the initial 7-month period from autumn to spring, DNA degradation occurred during the additional 8-month storage at RT from spring to autumn. This suggests that even when preserving samples in DESS, storage in cooler conditions is recommended to minimize DNA degradation.

We then attempted nondestructive DNA extraction from 600 µL of the DESS supernatant, followed by PCR amplification. Using the Zo-trnK-2F/2R primer pair [71], which produces a PCR product longer than 2 kb PCR, no amplification was observed in any of the samples. Shorter DNA barcode regions may potentially be amplified by PCR; however, further verification is needed.


**Algae**


Seaweeds are traditionally preserved and archived as pressed specimens (i.e., pressing seaweeds immediately onto a herbarium sheet and allowing them to dry). While pressed specimens preserve their internal morphology, they become flattened, so parts of the thallus are often preserved in formalin to maintain their three-dimensional internal structure. However, both pressed and formalin specimens are not suitable for DNA preservation. The most common method of DNA preservation is using a silica gel to rapidly dry parts of the thallus (e.g., [72,73,74,75]). It may not be examined how long such silica gel-dried specimens can preserve DNA, but PCR-quality DNA can be extracted from them even after at least 15 years of storage at RT (personal communication). Although isolating high-quality DNA from seaweeds is notoriously difficult due to the abundant polysaccharides and polyphenols in their tissues [76], PCR-quality DNA can be extracted using commercial kits with/without slight modifications in many cases (e.g., [77,78,79]). However, extracting large quantities of high-quality, high molecular weight DNA, which has also become increasingly in demand for seaweeds with the recent advent of whole-genome sequencing (WGS), remains problematic. The extraction of WGS-quality DNA depends not only on the method used but also on the condition of the sample. To extract WGS-quality genomes, it is common to use either freshly collected samples or those that have been immediately frozen in liquid nitrogen after collection and stored in a freezer. Silica gel-dried specimens are rarely used for extracting a WGS-quality genome. Silica desiccation may be problematic due to water stress and cell damage, promoting DNA damage, and it has also been suggested that polyphenols might chemically bind to DNA during desiccation, potentially leading to sequencing inhibition. Ref. [80] reported sequencing inhibition and reduced sequencing output when using silica gel-dried specimens for Oxford Nanopore sequencing.

We preserved *Turbinaria ornata*, a brown alga known for its difficult DNA extraction, in DESS for 3 months to examine both morphological and DNA preservation in the specimen. While DESS became brown-colored, no discoloration or external morphological changes were observed in *T. ornata* (Figure 26a). For DNA analysis, using approximately 0.1 g wet weight (Figure 26b), we first attempted DNA extraction using the modified Boom’s method (see Section 2.2.2. DNA extraction, Method 3), which yielded only 1.1 ng/µL (total 11 ng) of DNA, making it impossible to evaluate the genomic DNA quality. Subsequently, when we employed a DNA extraction method routinely used for brown algae ([81,82], see Section 2.2.2. DNA extraction, Method 4), we successfully extracted 18.9 ng/µL (200 ng) of DNA. Quality assessment using TapeStation (Agilent) revealed that relatively high molecular weight DNA was preserved, with a DIN value of 5.3 and a peak at 26.3 kb (Figure 26c,d). Although we need to investigate temporal changes in DNA quality because we did not check the DNA condition immediately after collection, we confirmed the effectiveness of DESS by obtaining DNA of sufficient quantity and quality for next-generation sequencing library preparation from specimens preserved at RT for three months. Verifying its effectiveness across various algal species in future studies could demonstrate its potential utility in genomics research.

#### 3.2.4. Fungi

Compared with some animal groups, DESS has not been used extensively for fungal materials. More frequently, other DNA extraction buffers, such as CTAB buffer, have been used for storing mushroom tissues for subsequent DNA studies (e.g., [83]). Some limited examples using DESS (including some modified versions of the original DESS) include the studies by Hosaka and Castellano [23], Hosaka [84], Hosaka and Uno [24], and Hosaka et al. [85].

Hosaka and Castellano’s [23] first study used a modified DESS, which includes the original DESS with the addition of 100 mM Tris-HCl (pH 8.0) and 0.1 M sodium sulfite. Before this study, one of the authors (KH) had routinely used the DESS with Tris but without sodium sulfite. However, some mushroom tissues showed significant discoloration (mostly becoming brownish) after storage for a few months at 4 °C (KH, personal observation). It was presumably (but not exclusively) due to the oxidation process. Therefore, to prevent potential oxidation, 0.1 M sodium sulfite was added in the subsequent experiments following Byrne et al. [86].

With the use of a modified DESS buffer, no significant discoloration of mushroom tissues was observed (Figure 27a), and most DNA experiments using this buffer have been successful from the samples collected more than 10 years ago (Figure 27b). Hosaka and Uno (2011) [24] examined the effect of the drying temperature and storage conditions on DNA in mushroom specimens and concluded that the DNA quality in the tissues preserved in a modified DESS buffer was not significantly different from that stored at −80 °C. It was also demonstrated that the DNA of mushroom tissues stored at RT in tropical countries was preserved in good quality by Hosaka et al. [85].

Overall, the quality and quantity of extracted DNA were higher from the tissues stored in DESS-NMNS buffer for one month (Figure 27b,d, “Control”). The tissues stored in DESS-NMNS for 15 years also yielded good quality/quantity DNA, but some longer amplifications did not produce clear bands (Figure 27b “DESS-NMNS”). This assessment was consistent with the results obtained using the TapeStation (Figure 27d). The quality/quantity of DNA from the dried specimens (15 years old) was the lowest in all experiments. The ITS region (ca. 600 bp) was successfully amplified for all samples, but some produced very weak bands, and long PCR produced only faint bands or no bands at all (Figure 27b, “Dry specimen”). The TapeStation also showed no visible bands for high bp. Regions, which are visible both for “Control” and “DESS-NMNS”, indicate that the quality of DNA from the 15-year-old dried specimens is low and the genomic DNA molecules are more fragmented (Figure 27d).

However, these results should be interpreted with caution. The low quality/quantity of DESS stored tissues does not necessarily directly indicate the situation in the tissues. Because of the long-term storage in high salt concentrations, the efficiency of DNA extraction may have been affected. In addition, the amount of tissue used for each DNA extraction was not strictly standardized. The low quality/quantity of DNA from dried specimens should also be considered with caution. It is a well-known fact that DNA in specimens degrades over time, but 15 years should not generally be considered a long time. Previously, our fungal herbarium had been regularly fumigated with methyl bromide, which is known to effectively fragment DNA molecules. However, the specimens used in this study have only been treated with sulfuryl fluoride, which arguably does not affect the quality of DNA [87]. We therefore conclude that although the quality/quantity of DNA from long-stored tissues is lower, the protocol for DNA extraction should be adjusted according to the type of tissues and storage methods.

### 3.3. Customizing the Preservation Solutions for Different Species

As evident from the above results with various species, appropriate preservation conditions differ depending on the organism. While DESS can be utilized as a tissue preservation solution for DNA across all species, as used in nematodes, it is necessary to adopt preservation solutions tailored to each species to maintain not only DNA in tissues but also taxonomic morphology in good conditions at RT over the long term. The quickest way to select appropriate preservation solutions for each species is for specialists in the target biological group to test each preservation solution. In some cases, individual researchers may need to customize solutions. Therefore, based on our verification, we itemized the criteria for selecting and customizing preservation solutions, conditions with room for customization, and points that should be verified in the future. In addition, please refer to the table of reagents and specimen processing methods (Table 3). For samples preserved for more than 10 years under any condition, further verification is necessary.

#### 3.3.1. Cost-Based Selection of Preservation Solutions

When selecting preservation solutions for biological specimens, cost becomes a crucial factor, especially for large-scale sampling projects or long-term collections. Since DNA analysis becomes difficult when formaldehyde is used for specimen preparation, it is desirable to preserve a portion of the tissue for DNA analysis. As shown in Table 4, the cost of the preservation solutions varies widely from approximately 50–100 JPY per 100 mL for basic ethanol solutions to about 19,600 JPY for commercial preservation reagents.

The most cost-effective option is 99% industrial ethanol, which is relatively easy to obtain and costs about 2 JPY per tube. Its volatility means it does not leave residue on specimens, making it user-friendly. However, this volatility becomes a disadvantage for long-term storage. 70% ethanol, commonly used for morphological preservation in animal specimens, is the most economical option at just 1.5 JPY per tube. It has proven effective for preserving morphological features over many years. For stable preservation of both morphology and DNA, 70% ethanol with EDTA (pH 8.0) added to a final concentration of 10 mM can be used at 2.4 JPY per tube, offering enhanced preservation performance with minimal cost increase.

Propylene glycol, at 10 JPY per tube, is also relatively cost-effective. Its nontoxic nature and chemical stability make it a useful option for insects and other organisms where dehydration does not alter morphology. However, since their dehydration capacity decreases with the water content of the samples, the specimens must be pretreated with ethanol for complete dehydration before preservation in propylene glycol.

DESS costs 25 JPY per tube, which is considerably more expensive than ethanol solutions, but its DNA preservation capability for long-term storage is significantly superior. Its modified version, DESS-NMNS, with added pH stability and antioxidant properties, is available at 29 JPY per tube. It offers high DNA degradation inhibition and protection against light-induced DNA damage for long-term RT DNA preservation, while also maintaining morphology in species where dehydration compromises taxonomic features. On the other hand, because DMSO has cytotoxicity, the PESS (propylene glycol/EDTA/saturated salt) preservation solution, which substitutes DMSO with propylene glycol, can be used when specimens can be stored in complete darkness. At 22 JPY per tube, it is slightly less expensive than DESS and offers improved safety, although it may lack the DNA photoprotection function of DMSO.

RNAlater, a commercial tissue preservation solution with ammonium sulfate as its main component, is extremely expensive at 196 JPY per tube. While it is possible to prepare a homemade version of RNAlater [88], its performance as a DNA preservation solution is not particularly high due to its acidic nature. Furthermore, due to ammonium sulfate’s property of removing bound hydration water from the protein, it is often unsuitable for maintaining taxonomic morphology, offering few significant advantages as a specimen preservation solution.

The ideal approach is to strategically select preservation solutions based on the value and intended use of the specimens. For general specimen collections intended only for morphological research, 70–80% ethanol provides sufficient preservation quality while being the most economical. For specimens also intended for DNA analysis, 70% ethanol + EDTA offers excellent cost-effectiveness with a minimal cost increase. For particularly valuable specimens or those requiring long-term DNA preservation, DESS or DESS-NMNS is preferable. While these solutions have higher initial costs, they offer high cost-effectiveness in terms of long-term protection of the scientific value of the specimens.

For large collections with multiple purposes, a tiered approach using different preservation solutions according to the specimen type and importance is the most economical. For example, using only ethanol or 70% ethanol + EDTA for the majority of specimens, while using DESS only for particularly important specimens or type specimens. In conclusion, while cost is an important factor in selecting preservation solutions, the balance between preservation objectives and specimen value should always be considered. Although low cost options can provide sufficient preservation quality when used appropriately, for specimens of high value, using solutions like DESS is important to maximize long-term scientific value, despite the additional cost.

#### 3.3.2. Safe Use of Preservation Solutions

DESS is a versatile preservation solution, but it requires careful handling as it contains DMSO, which has low toxicity, and EDTA at high concentrations, which has not been reported to affect human health but has a high environmental impact. In museums, when the purpose is to preserve specimens or tissues, disposal of the DESS solution is not anticipated, but if the need to dispose of the DESS arises, the waste must be treated as industrial waste.

DMSO is widely present in nature and is widely used as a useful organic solvent in various industries, but it shows strong permeability to biological cell tissues. Although DMSO itself has low toxicity to cells, its strong permeability may carry harmful substances into the body through the skin. Propylene glycol also shows similar strong permeability, so the use of the neat solution, such as in specimen preparation, requires similar caution. There are products that contain propylene glycol at about 25% in disinfectants, and if the content is 20% as in PESS, the risk is low. It has also long been known to show antibacterial activity against Gram-negative bacteria [89], and inhibitory or lethal effects specifically on Gram-negative bacteria such as *E. coli* have been confirmed at concentrations of 10% or more [90]. However, since it does not affect Gram-positive bacteria or fungi, using propylene glycol in its neat form can prevent the decay of biological specimens by lowering water activity through its dehydration capability. In PESS, the concentration of propylene glycol is 20%, but it contains 250 mM EDTA and saturated salt water, and EDTA is known to enhance the sensitivity of antimicrobial agents [91,92]. In addition, saturated NaCl can prevent the decay of biological specimens by lowering the water activity. If preservation for decades to hundreds of years is anticipated (where the possibility of disposal is extremely low), DESS or DESS-NMNS should be used, while for preservation not planned for such long terms, propylene glycol alone or PESS may be easier to use as these solutions also do not evaporate like ethanol and therefore require no maintenance for replacement due to evaporation.

#### 3.3.3. Application in Ecological Field Studies

Based on the insect test results (Section: Insects, Figure 10b), even low concentrations such as 5% DESS demonstrated high DNA preservation efficacy, suggesting promising applications for efficient field use in insect collection. For example, researchers can carry 50 mL of DESS to the collection site and dilute it 20-fold with water on-site to create 1 L of 5% DESS, dramatically reducing the load to carry. For collection periods ranging from several days to a few weeks, using even lower concentrations of DESS would likely be sufficient for DNA barcoding analysis. Furthermore, the collected insects can be gathered using nets and temporarily stored together in a bottle of 100% DESS during transport, then separated into individual specimens in the laboratory for individual DNA barcoding (Figure 11). The application range is expected to extend to spiders and other organisms, warranting further verification in future studies.

However, caution regarding the environmental impact is necessary when using DESS in the field. DMSO is known to adversely affect cellular functions even at low concentrations of 0.1%. According to toxicity data on aquatic organisms studied by the Ministry of the Environment, preliminary reference data indicate that DESS diluted 440-fold affected the reproduction of water fleas (*Daphnia magna*) [93]. EDTA, although used in cosmetics and detergents, has extremely low biodegradability and places a significant burden on the environment. Therefore, for field studies with risks of environmental release, it is desirable not only to replace DMSO with propylene glycol but also to substitute EDTA with a less environmentally impactful chelating agent. For instance, biodegradable chelating agents (HIDS^®^, Nippon Shokubai, Co., Ltd.) used in laundry detergents, soil conditioners, and cosmetic ingredients might be suitable replacements for EDTA. When HIDS^®^, which is a pH 10.3 liquid, was adjusted to pH 8 using 1N-HCl and tested for DNase inhibitory effect, it demonstrated an equivalent DNase inhibitory effect to 250 mM EDTA at a concentration of approximately 5% (Figure 28). Replacing the 250 mM EDTA in DESS with 5% HIDS^®^ to create PH-SS (P.G./HIDS^®^/Saturated Salt) could produce a preservation solution with reduced environmental impact. In ecological research, with a high risk of environmental release, diluting PH-SS in the field (e.g., 20-fold dilution to 5% or less) could maintain DNA preservation while reducing the environmental impact, potentially contributing to the development of safer ecological experimental systems.

### 3.4. DNA Extraction from Museum Specimens

When extracting DNA from specimens or tissue preserved in DESS, the tissue fragments contain large amounts of salts (EDTA and NaCl), which are PCR inhibitors. Therefore, thorough desalination is necessary during the DNA extraction process. For example, washing twice with 70–75% ethanol at the end of purification is effective. In PCR, even if the DNA concentration is quite low, amplification of the DNA barcoding regions often succeeds as long as PCR inhibitors are not present. Therefore, we recommend thorough washing at the final step of extraction.

DNA extracted from samples preserved in DMSO-containing DESS is reported to undergo methylation [18]. However, DMSO methylates DNA and RNA in the presence of hydrogen peroxide and iron catalysts [94], so the possibility of methylation being promoted in DESS containing high concentrations of EDTA is low, and DESS itself does not have the function to methylate DNA. Nevertheless, insects sometimes remain alive for several minutes after being placed in DESS, and since cells do not die immediately when specimens are placed in DESS, intense stress responses are likely occurring within the cells during this time. Therefore, for insects, it is recommended to kill them instantly with ethanol or ethyl acetate before preserving them in DESS. If DNA is to be extracted relatively soon after collection, dry, refrigerated, or frozen storage might be better.

Although not included as test materials in this study, for land plants, which naturally have less moisture content, preservation by drying or freezing is most suitable. To safely preserve DNA, it is advisable to take a portion of tissue into an experimental tube, dry it using a freeze dryer ensuring it is not exposed to gradually increasing heat, remove it once it reaches RT, immediately close the tube lid, and store at RT, or ideally under refrigeration or in a freezer. In such cases, 1.5 mL or 2 mL tubes commonly used in molecular biology experiments are preferable to screw-cap storage tubes because of their higher air tightness. DNA can be extracted from completely dried samples or specimens, but since DNA degradation is accelerated by moisture in the air, heat drying at 65 °C for 30 min is also an effective method to irreversibly inhibit DNase for samples or specimens where color changes are not problematic.

#### Comparison of Silica Performance and Cost in DNA Extraction Using the Boom Method

DNA barcoding using the crude extract from specimen samples is a means to obtain sequence information at low cost, and various methods are utilized. However, crude extracts containing large amounts of contaminants have poor stability and can only be used as PCR templates for a short period. For specimens registered as collections, it is desirable to preserve their DNA, requiring the extraction of clean DNA free from contaminants at a low cost. Additionally, depending on the tissue, DNA may be fragmented or present in very low concentrations, requiring the selection of DNA extraction methods based on the sample conditions. As described in the Methods Section, the NMNS frequently employs the Boom method because of its excellent ability to remove contaminants during DNA extraction and its cost-effectiveness (refer to Methods). The Boom method [26] is a DNA extraction technique that utilizes the property of nucleic acids to adsorb to silica beads in the presence of chaotropic salts Vogelstein and Gillespie, 1979 [95]. This method is also employed in various DNA extraction kits. With this technique, by preparing silica and chaotropic agents, it is possible not only to set up DNA extraction experimental systems tailored to sample volumes but also to easily wash away contaminants by exploiting the property of nucleic acids to adsorb to silica.

However, even with methods based on the same principle, the results can differ depending on how they are implemented. For example, among DNA extraction kits, it has been reported that column-type extraction using silica membranes yields higher DNA quantities but more contaminants, while silica-coated magnetic beads yield lower DNA quantities but fewer contaminants Rothe and Nagy, 2016 [96]. This indicates the necessity of selecting methods according to the experimental objectives and materials.

The Boom method commonly employs chaotropic agents such as guanidine hydrochloride and sodium iodide, which were also used in this study. On the other hand, although various types of silica are used, there has been limited discussion regarding their DNA extraction efficiency and cost. Therefore, we conducted a comparative experiment using the following four types of silica utilized within the National Museum of Nature and Science (Table 5) to perform a simplified investigation of the properties and the DNA extraction efficiency.

First, we prepared silica suspensions (Glass milk) for DNA adsorption. When measuring approximately equal amounts of dry powder using an electronic balance, we observed that despite the silica powders having almost identical weights, they differed in appearance and seemed to vary slightly in volume (Figure 29a). After adding water at 10 times the weight of silica and allowing it to settle, we observed clear differences in the sedimentation rates of the silica particles (Figure 29b–d). For Silica gel (Davisil 643 amorphous silica particles, Cat. No. 236810; Sigma-Aldrich, St. Louis, MI, USA), most particles had settled after just one minute, while only a portion of the diatomaceous earth (Cat. No. D3877; Sigma-Aldrich, St. Louis, MI, USA) had settled, and almost no sedimentation was observed for Wakosil (Cat. No. 239-00851; FUJIFILM, Japan) and SiO₂ (Cat. No. S5631; Sigma-Aldrich, St. Louis, MI, USA). Suspecting these differences were due to variations in silica particle size, we conducted electron microscope observations (Figure 30). We found that Silica gel consisted of very large particles, diatomaceous earth contained particles of various sizes, Wakosil had particles of uniform size, and SiO₂ was composed of the smallest particles with varying dimensions, which explained the differences in sedimentation rates in water. These size differences in silica particles create variations in surface area available for DNA adsorption, suggesting that DNA extraction efficiency might differ among the silica types.

To investigate the extraction efficiency of high and low molecular weight DNA, we used genomic DNA extracted from bird breast muscle and a 100 bp DNA ladder (Takara). After adding an equal volume of AL Buffer (QIAGEN) to the DNA, we added 10 μL or 25 μL of silica suspension to adsorb the DNA, then purified it by washing with 75% ethanol. We found that Wakosil and Silica gel failed to purify either genomic DNA or ladder DNA, whereas SiO_2_ and diatomaceous earth recovered DNA at similar levels (Figure 31a,b). There was no correlation between the amount of silica suspension and DNA recovery; rather, with 25 μL of SiO_2_, the larger amount of silica required more time to dry the ethanol completely, resulting in failure to recover DNA in some samples (Figure 31b, SiO_2_-1). Silica gel becomes transparent when it contains moisture, making the precipitate difficult to see and requiring extreme care not to aspirate the silica when discarding the supernatant. To promote DNA adsorption to Wakosil and silica gel, we added an equal volume of ethanol to the DNA solution and AL buffer before adding the silica suspension, which improved the DNA recovery rate (Figure 32). When using Wakosil and silica gel for DNA extraction, it appears necessary to add ethanol in addition to chaotropic agents to adsorb DNA to the silica.

The glass milk is used after being suspended in either water or 0.01 N HCl, though we have not obtained clear differences in DNA extraction amounts between these two options (Figure 31). However, when suspended in water, there is a risk of deterioration, so except when using it up in a short period, preparing it with 0.01 N HCl might be safer. This is supported by the fact that in the Boom method, slightly acidic conditions are considered favorable for DNA adsorption to silica [97,98]. Tissues preserved in DESS are alkaline (pH 8.0), which could shift the silica adsorption conditions toward alkalinity during DNA extraction. While this raised concerns about decreased DNA extraction efficiency, we conducted verification experiments performing silica adsorption under alkaline conditions by adding 30 μL of DESS, and found no difference in the amount of DNA recovered after purification with any of the silica types. Under the conditions using 500 ng of DNA that we employed in this study, it seems unnecessary to be overly concerned about the pH increase caused by DESS in the tissue.

Finally, we compared the costs of the silica types (Table 5). There was a 200-fold price difference between the most expensive Wakosil and the least expensive diatomaceous earth. While further verification with more minute amounts of DNA is necessary regarding the DNA extraction efficiency of each silica type, inexpensive diatomaceous earth or silicon dioxide should be sufficient for simple DNA analysis, such as DNA barcoding. Silica gel becomes glass-like and transparent in water, making the pellet difficult to see and handle. Wakosil, with its highly uniform particles, provides high reproducibility. It seems best to select the appropriate silica type according to the specific purpose.

**Table 5 biology-14-00730-t005:** Comparison of the silica cost and performance.

Product Name	Cost Per 10 g (Dollar, JPY)
Wakosil^®^5SIL ^1^	USD0.014, JPY23,800
Silica gel ^2^	USD4.73–10.95, JPY687–1590
SiO_2_ (~99%, 0.5–10 μm) ^3^	USD1.42–7.44, JPY206–1080
Diatomaceous earth ^4^	USD0.81–2.04, JPY117–296

Costs are shown in Japanese JPY (¥) as of January 2025, with USD equivalents calculated using an exchange rate of JPY145.23 = USD1.00. Dollar values are based on reagent prices in Japan. ^1^ Wakosil^®^5SIL, Cat. No. 239-00851; FUJIFILM, Japan, calculated from 10 g [99]. ^2^ Davisil 643 amorphous silica particles (Cat. No. 236810; Sigma-Aldrich, St. Louis, MI, USA, calculated from 100 g and 1 kg [100]. ^3^ Cat. No. S5631; Sigma-Aldrich, St. Louis, MI, USA, Calculated from 100 g and 1 kg [101]. ^4^ Diatomaceous earth, Cat. No. D3877; Sigma-Aldrich, St. Louis, MI, USA, calculated from 500 g and 5 kg [102].

## 4. Conclusions

Our preliminary experiments demonstrated that DESS effectively preserved DNA while maintaining morphological integrity across diverse organisms, including nematodes, spiders, insects, shrimp, seagrasses, and algae. Even at reduced concentrations, DESS exhibited sufficient DNA preservation effects, although further validation through long-term testing across various species remains necessary. It is important to note that the EDTA component in DESS dissolves calcareous structures, rendering it unsuitable for the morphological preservation of sea cucumbers and shellfish; in these cases, DESS should be used exclusively for the DNA preservation of sea cucumber tissues and only for preserving the soft body parts of shellfish after shell removal.

DESS enables long-term tissue preservation of DNA at RT. Our experiments with chicken muscle tissue revealed no DNA degradation even after 2 years of storage at RT in 70% DESS, suggesting that 100% DESS may not be strictly necessary. However, determining the optimal concentration that balances preservation efficacy, practical convenience, and cost-effectiveness requires further investigation. The superior tissue penetration properties of DESS compared with ethanol and formalin were clearly demonstrated in our experiments with chicken muscle tissue. DMSO facilitates the distribution of EDTA and NaCl throughout the tissue interior, where a high salt concentration suppresses tissue decay while EDTA protects DNA.

For chicken muscle tissue preservation, DESS exhibited remarkable stability even during preservative evaporation, with DNA quality maintained in tissues after complete evaporation, unlike ethanol preservation, where DNA degraded rapidly under similar conditions. This stability indicates DESS’s potential for institutions where preservation spanning decades to over a century is anticipated. In addition, DESS has demonstrated versatility across biological kingdoms, preserving DNA in a wide range of organisms.

From a cost perspective, while DESS (25 JPY/tube) is more expensive than ethanol solutions (1.5–2.4 JPY/tube), its superior DNA preservation capabilities justify the additional expense for valuable specimens or those requiring long-term preservation. For organizations maintaining large collections, a tiered approach using different preservation solutions according to specimen importance represents the most economical strategy.

For safer field applications with reduced environmental impact, our research suggests promising alternatives such as replacing DMSO with propylene glycol and EDTA with biodegradable chelating agents like HIDS^®^. Notably, diluted DESS solutions (as low as 5%) demonstrated effective DNA preservation in insects, suggesting the possibility of reducing both the cost and the environmental impact during field collection by on-site dilution.

## 5. Future Perspectives

In conclusion, while DESS represents an excellent solution for long-term DNA preservation across diverse organisms, tailoring preservation methods to specific taxonomic groups and research objectives remains essential. Future research should focus on optimizing DESS concentrations, exploring environmentally friendly alternatives, and conducting extended preservation tests across a broader range of species to fully establish the long-term reliability of these preservation methods across the biological spectrum. Additionally, integrating this preservation approach with emerging technologies such as microfluidic systems could enable automated and high-throughput processing of specimens in both field and laboratory settings. The implications for biodiversity research and taxonomic studies are particularly promising, as improved preservation methods could accelerate the discovery and documentation of new species while maintaining valuable reference collections. Furthermore, standardizing these optimized preservation protocols across scientific institutions would enhance collaborative research efforts and contribute to global biodiversity initiatives.

Until now, DNA analysis of museum specimens has primarily relied on short-read sequencers (second generation sequencers) capable of analyzing minute DNA quantities. However, long-read sequencing (3rd generation sequencer) protocols that can function with as little as 1 ng of DNA have recently been published [103], suggesting that future specimen preservation should consider the potential for telomere-to-telomere (T2T) genome sequencing. Moving forward, it will also be necessary to verify whether DNA from DESS-preserved specimens can be utilized for Hi-C techniques [104,105] and whether the three-dimensional structure of the genome is maintained in these samples.

## Figures and Tables

**Figure 1 biology-14-00730-f001:**
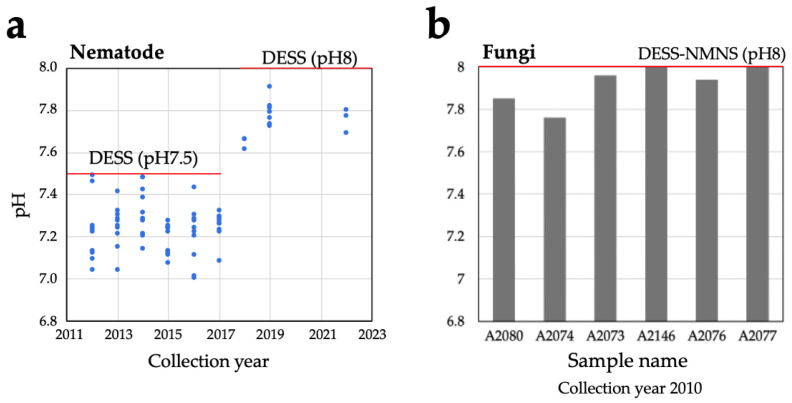
The pH of the DESS (DMSO, EDTA, and saturated salt) solution used for nematode and fungal specimen preservation at room temperature. (**a**) pH of DESS for specimens prepared from 2012 to 2022. DESS pH 7.5 was used from 2012 to 2017, and DESS pH 8.0 was used from 2018 onward. Each dot represents an individual specimen. The red line indicates the pH of DESS before sample immersion. (**b**) The pH of DESS-NMNS (DESS + Tris + Na_2_SO_3_) for specimens prepared from 2010. Each bar represents an individual specimen. The red line indicates the pH of the DESS before sample immersion.

**Figure 2 biology-14-00730-f002:**
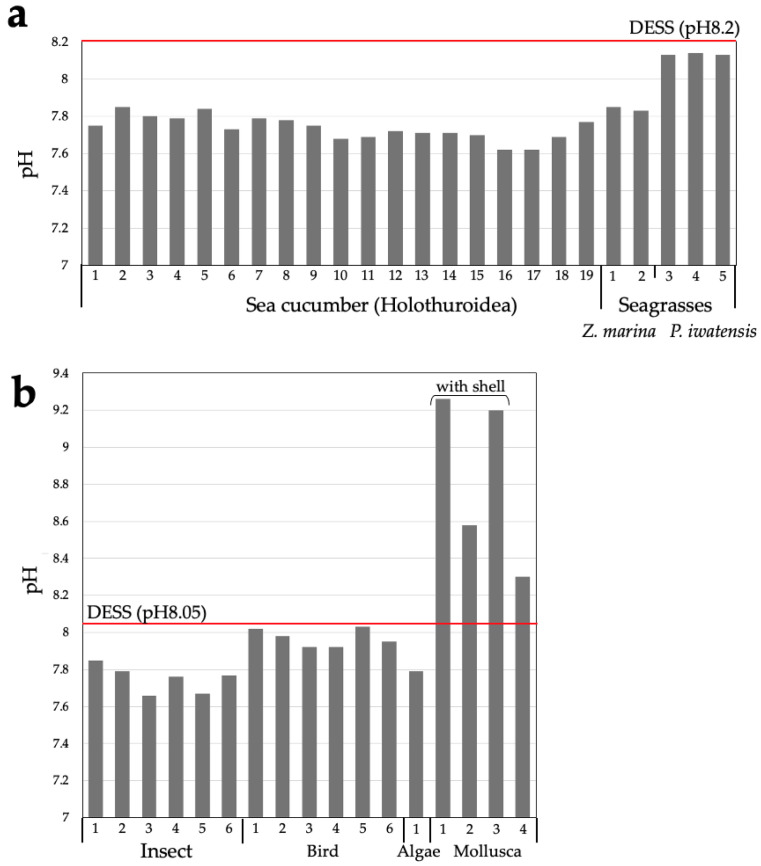
The pH of the DESS used for the preservation of various specimens. (**a**) The pH of the DESS used to preserve sea cucumber (Holothuroidea) specimens. DESS (pH 8.18) was used during this period. (**b**) The pH of the DESS used for the preservation of insect, seagrass (*Phyllospadix japonica*), seaweed (*Turbinaria ornata*), and gastropod (1: *Lottia kogamogai*, 2: *Echinolittorina radiata*, 3: *Sinotaia quadrata histrica* with shell, and 4: *Sinotaia quadrata histrica* without shell) specimens. DESS (pH 8.0) was used. Each bar represents an individual specimen.

**Figure 3 biology-14-00730-f003:**
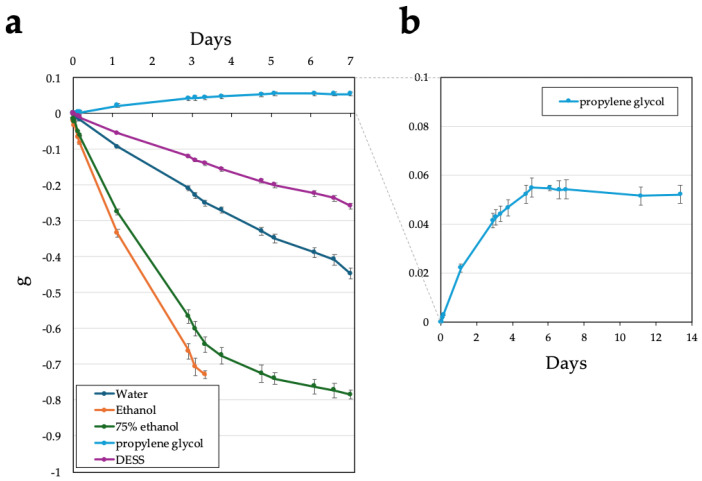
Comparison of the evaporation rates of the preservation solutions under open-air exposure at room temperature. Water (as control), anhydrous ethanol, 75% ethanol, DESS, and propylene glycol (1 mL each) were placed in 2 mL tubes (*n* = 3 each) and stored at room temperature with open lids, and their weight loss was measured. (**a**) Weight change for each solution over 7 days. (**b**) Weight change for propylene glycol over 14 days. Error bars represent standard error (SE).

**Figure 4 biology-14-00730-f004:**
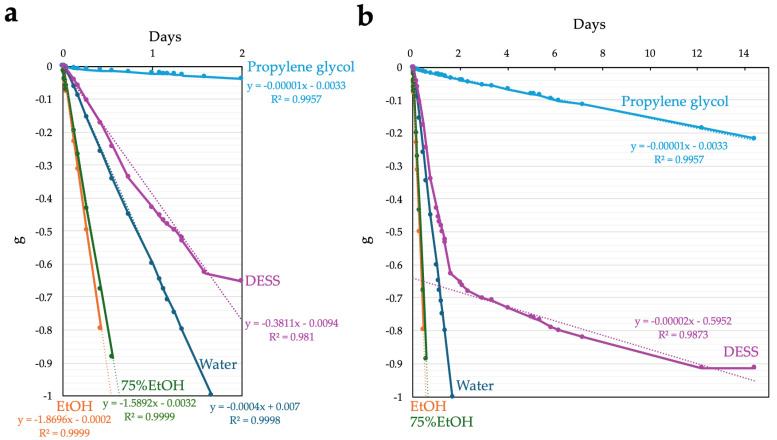
Comparison of the evaporation rates of the preservation solutions under open-air exposure at 55 °C. Water (as control), anhydrous ethanol, 75% ethanol, DESS, and propylene glycol (1 mL each) were placed in 2 mL tubes (*n* = 3 each) and stored at room temperature with open lids, and their average weight loss was measured. (**a**,**b**) Weight changes in each solution over 2 days (**a**) and 14 days (**b**). The regression equations are shown for each solution. For DESS, the regression equation was calculated based on the weight decrease during days 0–2 in (**a**) and days 2–14 in (**b**). R^2^ indicates the correlation coefficient.

**Figure 5 biology-14-00730-f005:**
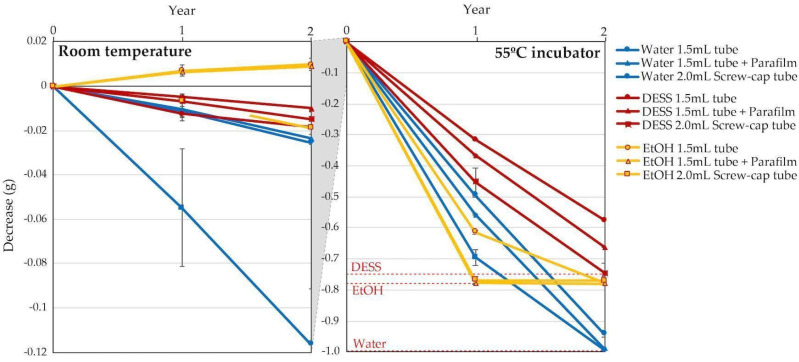
Differences in the evaporation rates of the preservation solutions under varying environmental conditions and tube types. Water (as control, 1 mL), DESS, or 99.5% (vol) ethanol (EtOH) was stored in 1.5 mL tubes (*n* = 3 each) and 2 mL screw-capped tubes (*n* = 3 each) at room temperature (**left**) and 55 °C (**right**), and evaporation was monitored over a 2-year period. An additional condition using Parafilm-sealed 1.5 mL tubes (*n* = 3) was also tested. The blue, red, and orange lines represent water, DESS, and EtOH, respectively. The red dashed horizontal lines indicate the complete evaporation amount for DESS, EtOH, and water, respectively. Error bars represent SE.

**Figure 6 biology-14-00730-f006:**
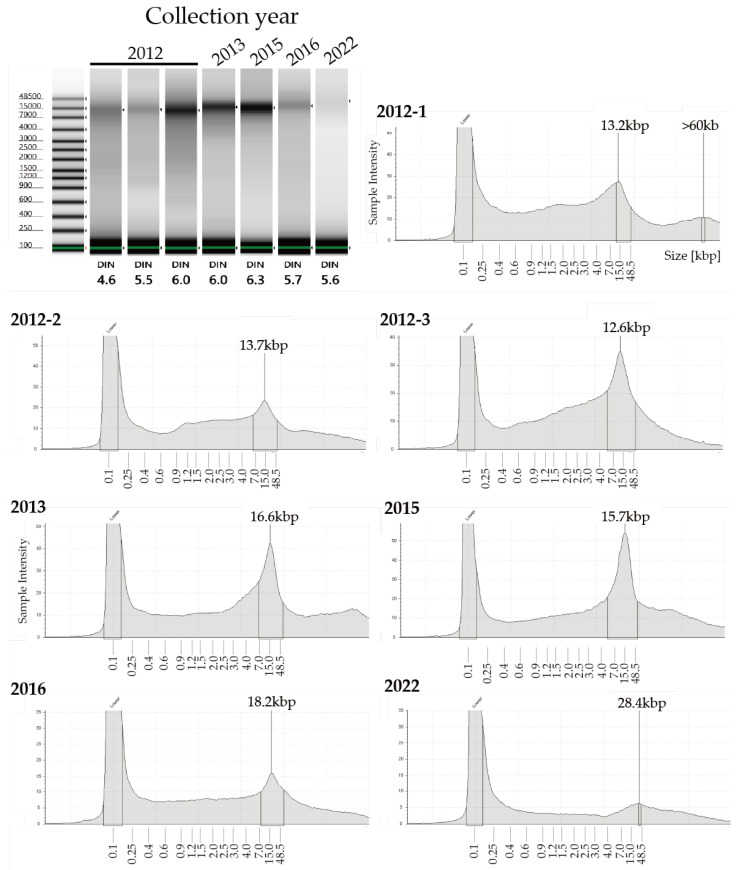
Gel image and electropherograms of extracted DNA from nematode specimens collected from 2012 to 2022. DNA was analyzed using an Agilent TapeStation 4400 system (Agilent Technologies) and the Agilent Genomic DNA ScreenTape assay to determine the DIN (DNA Integrity Number). The DIN was calculated using TapeStation analysis software based on fragmentation; its typical range is 0–10. A DIN value of 1 indicates highly degraded DNA, whereas a value of 10 represents completely intact genomic DNA. All samples showed an alert signifying a sample concentration that was outside the recommended range. The first peak at 100 bp is a molecular weight marker standard.

**Figure 7 biology-14-00730-f007:**
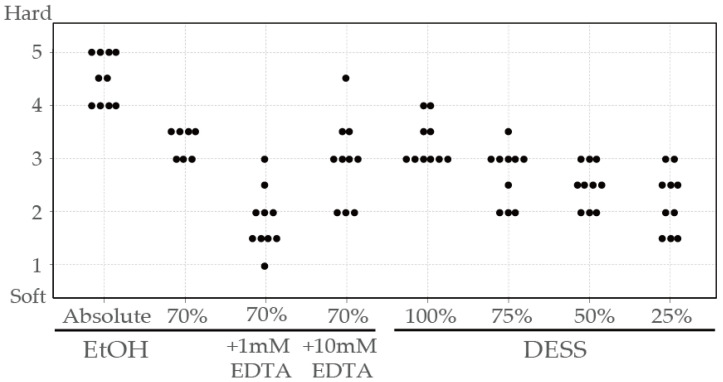
Relationship between the preservative solution composition and specimen rigidity for spiders. Subjective evaluation of specimen rigidity for spiders preserved in different solutions: anhydrous ethanol, 70% ethanol, 70% ethanol + 1 mM EDTA, 70% ethanol + 10 mM EDTA, DESS, 75% DESS, 50% DESS, and 25% DESS. Rigidity was assessed by handling specimens with forceps and scored on a scale of 1 to 5, where 1 indicated the softest and 5 indicated the hardest condition. Each dot represents an individual specimen.

**Figure 8 biology-14-00730-f008:**
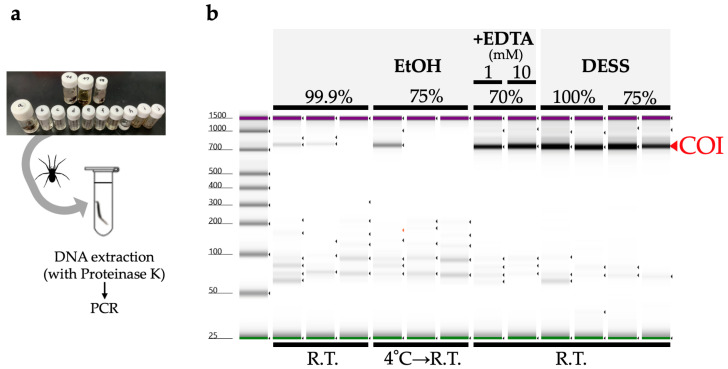
Comparison of DNA preservation efficacy among different preservative solutions for spider specimens. (**a**) Schematic illustration of the experimental workflow showing DNA extraction from the spider leg preserved under various conditions and subsequent PCR amplification of the COI (cytochrome c oxidase subunit I). (**b**) Gel image of the PCR-amplified COI products using DNA templates extracted from specimens preserved in various preservation solutions: anhydrous ethanol (99.9%), 75% ethanol, 70% ethanol containing either 1 mM EDTA or 10 mM EDTA (4 or 40 µL 500 mM EDTA (pH8) + 70% EtOH up to 2 mL), undiluted DESS, and 75% DESS. All specimens were preserved for 6 months (specimens in 75% EtOH were stored in a refrigerator for 2 months, followed by 4 months at room temperature (RT), while all other specimens were stored at RT. PCR products were analyzed using the Agilent TapeStation 4400 system and the Agilent D1000 ScreenTape assay (Agilent Technologies, Inc., Santa Clara, CA, USA).

**Figure 9 biology-14-00730-f009:**
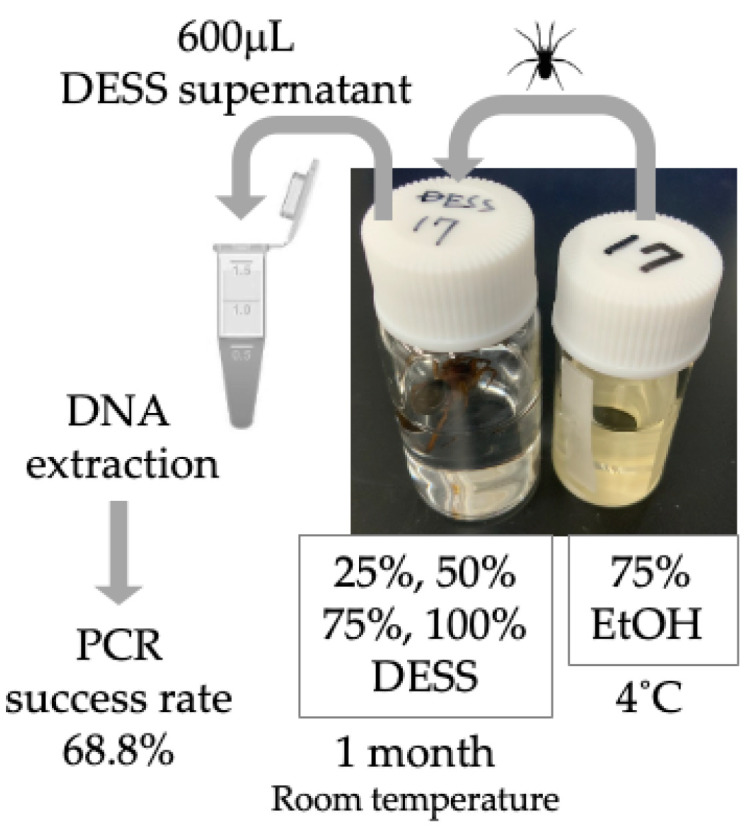
Schematic illustration of the experimental workflow for nondestructive DNA extraction from the DESS supernatant. Spider specimens were preserved in different DESS concentrations, with DNA extracted from a single leg per specimen for PCR amplification of the COI region. The glass vials containing the specimens had the following dimensions (inner mouth diameter × body diameter × total length): φ10 × φ21 × 45 mm [left] (9-852-05, LABORAN Screw Tube Bottle 9 mL, AS ONE Corp., Osaka, Japan) and φ10 × φ18 × 40 mm [right] (9-852-04, LABORAN Screw Tube Bottle 6 ml, AS ONE Corp.).

**Figure 10 biology-14-00730-f010:**
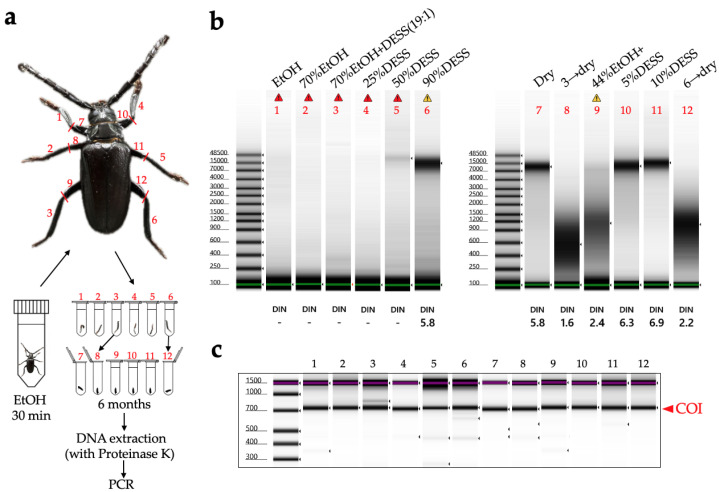
Comparison of DNA extracted from the tibiae and femora of *Prionus insularis* preserved under different conditions. (**a**) Schematic diagram of the experimental procedure. The specimen was dehydrated in anhydrous ethanol for 30 min before being transferred to preservation solutions under different conditions. Samples 1–6 used the tibiae while samples 7–12 used the femora. For samples 7 and 12, after 24 h in preservation solutions 3 and 6, respectively, excess liquid was absorbed with paper, and the specimens were air-dried in uncapped tubes. All tubes were stored in a laboratory drawer at RT under dark conditions. DNA extraction was performed after 6 months of storage. (**b**,**c**) Genomic DNA and PCR products were analyzed using the Agilent TapeStation 4400 system and the Agilent Genomic (**b**) or D1000 (**c**) ScreenTape assay. The DIN was calculated using TapeStation analysis software based on fragmentation; its typical range is 0–10. A DIN value of 1 indicates highly degraded DNA, whereas a value of 10 represents completely intact genomic DNA. The yellow alert indicates a sample concentration that is outside the recommended range. The red alert indicates DNA concentrations below the DIN value estimation limit. The first peak at 100 bp is a molecular weight marker standard.

**Figure 11 biology-14-00730-f011:**
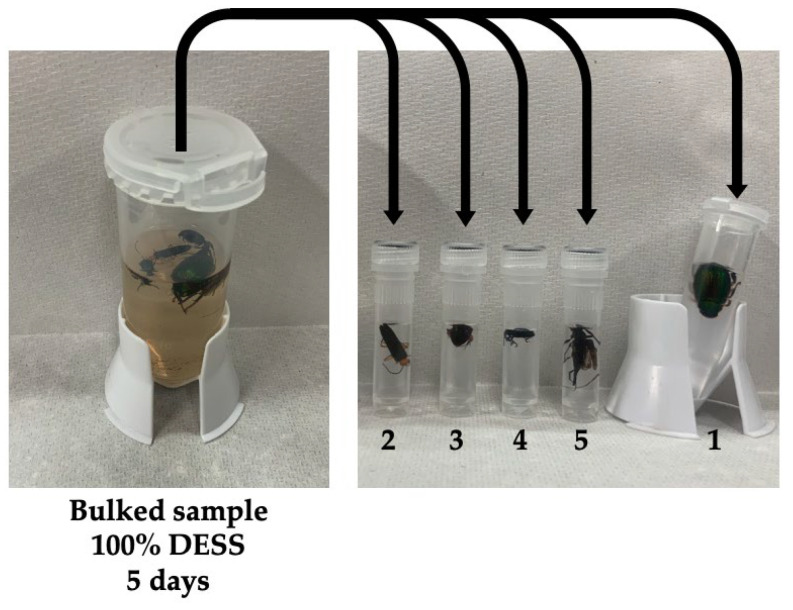
Verification of contamination during DNA extraction when preserving multiple insects in the same DESS solution.

**Figure 12 biology-14-00730-f012:**
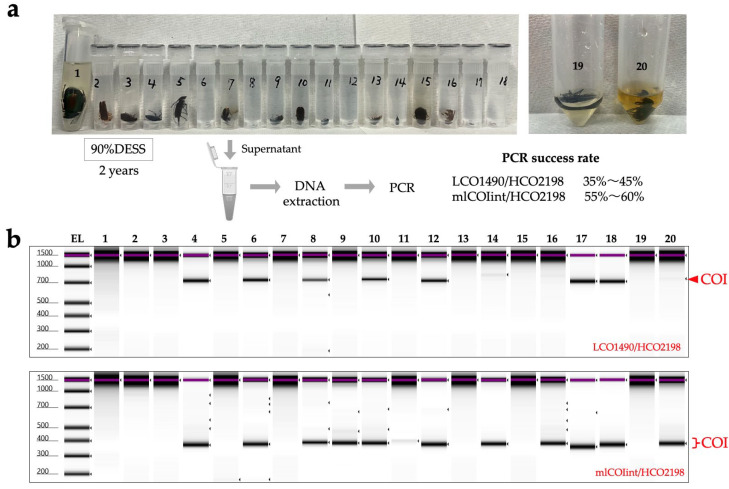
Nondestructive DNA extraction using the DESSS supernatant from various insects and COI PCR amplification. (**a**) Schematic illustration of the experimental workflow for nondestructive DNA extraction from the DESS supernatant. All specimens were preserved in 90% DESS at room temperature under dark conditions. After 2 years, 600 µL of supernatant was collected and used for DNA extraction. (**b**) Gel image of the PCR-amplified COI products (upper: LCO1490/HCO2198 [30]; lower: mlCOIint/HCO2198 [31] primer pair) using DNA templates extracted from the supernatant of 90% DESS-preserved specimens. PCR products were analyzed using the Agilent TapeStation 4400 system and the Agilent D1000 ScreenTape assay. EL indicates the electric ladder.

**Figure 13 biology-14-00730-f013:**
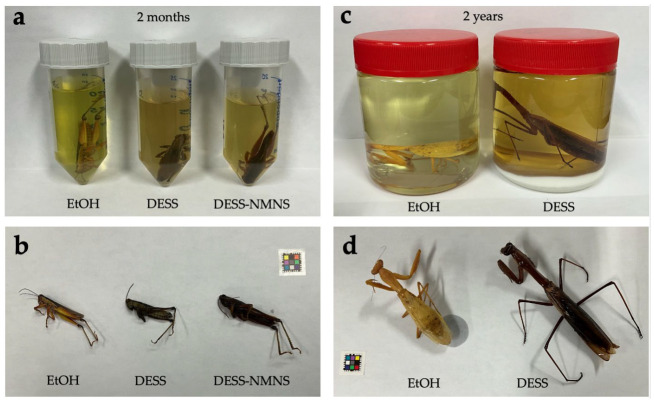
Color differences in Orthoptera insect specimens by preservation solution. (**a,b**) Color differences in specimens after 2 months of preservation in ethanol, DESS, and DESS-NMNS, respectively. (**c,d**) Color differences in specimens after 2 years of preservation in ethanol or DESS.

**Figure 14 biology-14-00730-f014:**
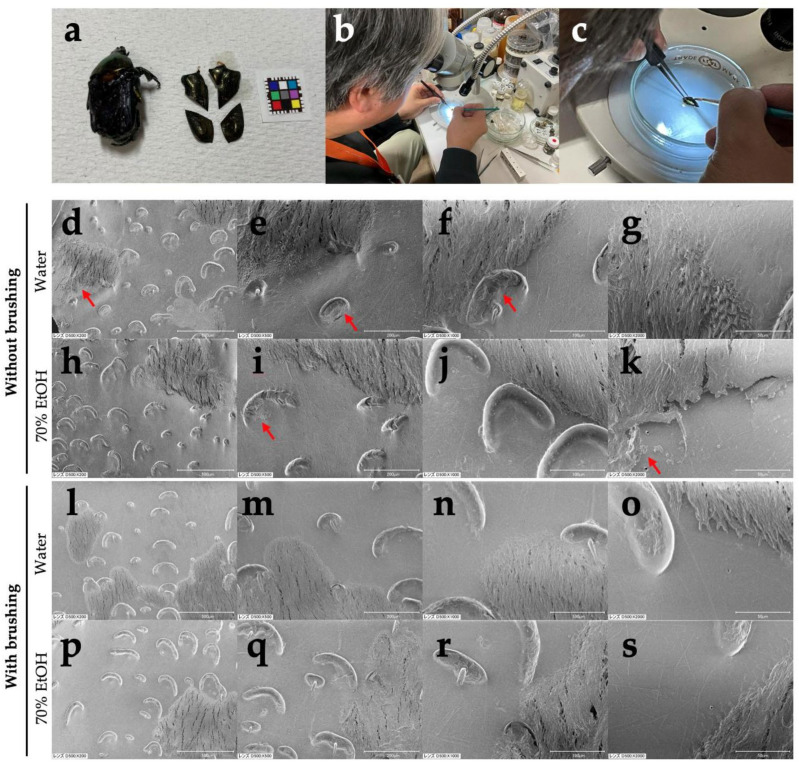
Desalination of DESS-preserved specimens (**a**) A oriental flower beetle (*Protaetia orientalis*) preserved in DESS for 1.5 years was removed from the solution, blotted with paper towels, and its wings were removed and divided into two parts, each with characteristic morphological features, resulting in a total of four wing sections. The left wings were placed in water, and the right wings in 70% ethanol, both gently stirred. The water and 70% ethanol volumes were approximately 100 times the weight of the wings (2 mL). (**b**,**c**) After 12 h, the upper parts of the cut wings were removed from the solution, blotted with paper towels, and air-dried. The lower parts of the cut wings were brushed on the surface using a fine brush under a microscope. (**d**–**k**) Surface structures of the *P. orientalis* wings without brushing. Wings were desalinated with either water (**d**–**g**) or 70% ethanol (**h**–**k**). (**l**–**s**) Surface structures of the *P. orientalis* wings with brushing. Wings were desalinated with either water (**l**–**o**) or 70% ethanol (**p**–**s**) and brushed in each solution. (**d**–**s**) Each electron microscope (KEYENCE VHX-D510) photograph shows the lens D500 and magnification in the lower left corner, at 200× (**d**,**h**,**l**,**p**), 500× (**e**,**i**,**m**,**q**), 1000× (**f**,**j**,**n**,**r**), and 2000× (**g**,**k**,**o**,**s**), respectively.

**Figure 15 biology-14-00730-f015:**
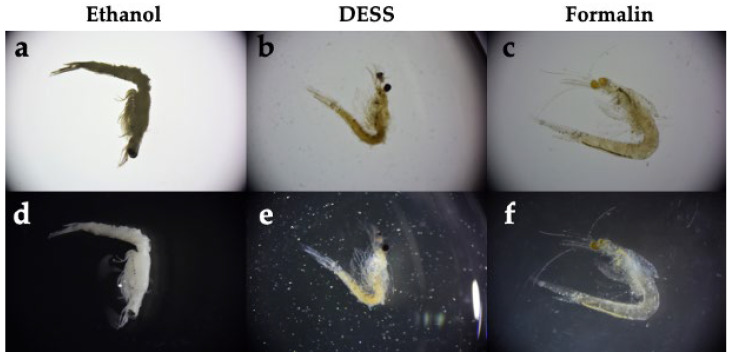
Comparison of preservation methods for *Neomysis japonica* specimens using different preservation solutions [ethanol, DESS, and formalin]. Specimens were photographed under light microscopy against (**a**–**c**) white and (**d**–**f**) black backgrounds to highlight the morphological differences. (**a**,**d**) Ethanol; (**b**,**e**) DESS; and (**c**,**f**) formalin.

**Figure 16 biology-14-00730-f016:**
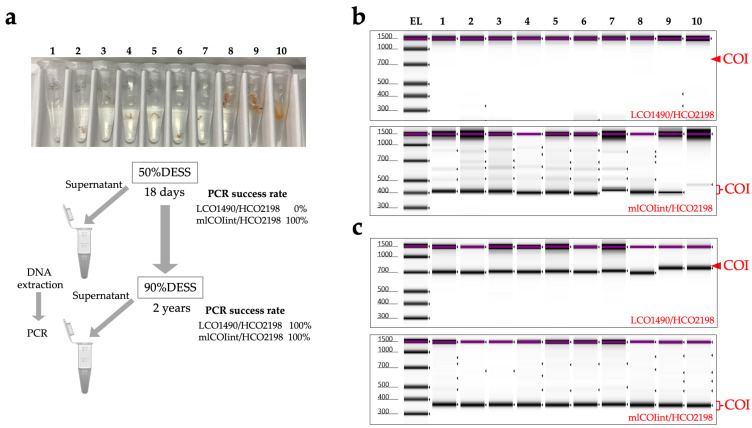
Nondestructive DNA extraction using DESS supernatant from *Neomysis japonica* and COI PCR amplification. (**a**) Schematic illustration of the experimental workflow for nondestructive DNA extraction from DESS supernatant. All *N. japonica* specimens were preserved for 18 days in 50% DESS at room temperature under dark conditions. After collecting 700 µL of supernatant for DNA extraction, 100% DESS was added to achieve a final concentration of approximately 90% DESS, and the specimens were preserved under the same conditions. After 2 years, 700 µL of supernatant was similarly collected and used for DNA extraction. (**b**,**c**) Gel image of the PCR-amplified COI products (upper: LCO1490/HCO2198 [30]; lower: mlCOIint/HCO2198 [31] primer pair) using DNA templates extracted from the supernatant of 50% DESS-(**b**) or 100% DESS-preserved specimens (**c**). PCR products were analyzed using the Agilent TapeStation 4400 system and the Agilent D1000 ScreenTape assay. EL indicates the electric ladder. The lower PCR products (mlCOIint/HCO2198) in (**b**) showed size irregularities because the samples had dried due to evaporation during refrigeration and were reconstituted with water before the TapeStation analysis.

**Figure 17 biology-14-00730-f017:**
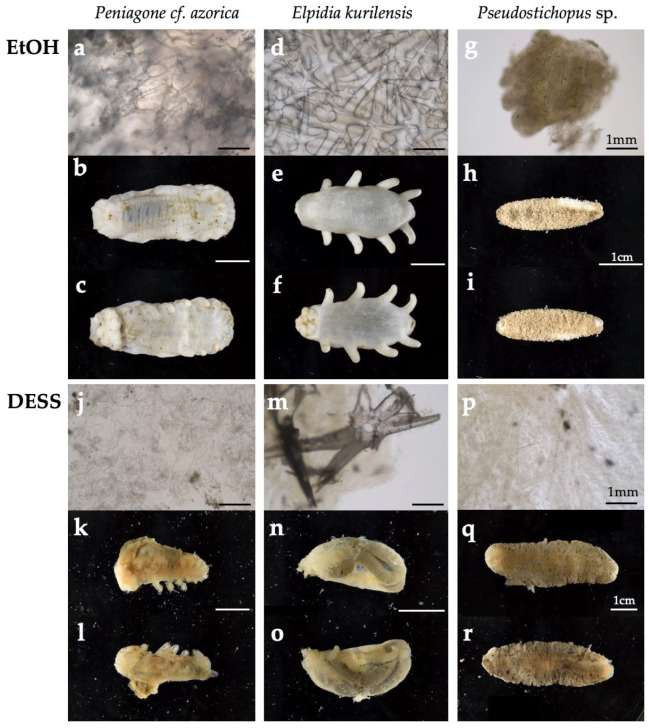
Comparison of preservation methods for three sea cucumber species ((**a**–**c**),(**j**–**l**) *Peniagone* sp.; (**d**–**f**),(**m**–**o**) *Elpidia kurilensis*; (**g**–**i**),(**p**–**r**): *Pseudostichopus* sp.) preserved in ethanol (**a**–**i**) and (**h**–**r**) DESS. Observation images for calcareous ossicles in body walls (**a**,**d**,**j**,**m**) and tentacles (**g**,**p**) under light microscopy. Whole body images of preserved specimens from the dorsal view (**b**,**e**,**h**,**k**,**q**), ventral view (**c**,**f**,**i**,**l**,**r**), left side view (**n**), and right side view (**o**).

**Figure 18 biology-14-00730-f018:**
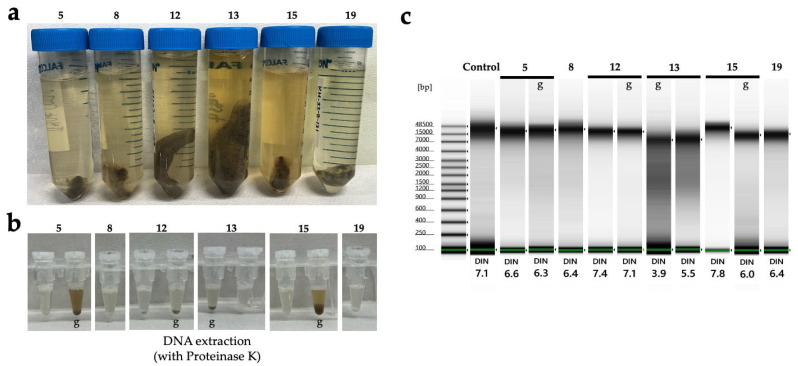
Comparison of sea cucumber (8, 15: *Peniagone* sp.; 5, 19: *Elpidia kurilensis*; 12, 13: *Pseudostichopus* sp.) DNA quality extracted from tissues preserved in ethanol or DESS. (**a**) Specimen collected at sea and immediately preserved in 40 mL of 100% DESS in a 50 mL tube for 26 months. (**b**) Tissue samples were taken with forceps from a specimen preserved in DESS for 26 months, undergoing Proteinase K treatment in ATL buffer. The samples included tissues of the tube feet or gut walls containing gut contents. Samples containing dark-colored gut contents are indicated as “g”. (**c**) Genomic DNA was analyzed using the Agilent TapeStation 4400 system and the Agilent Genomic ScreenTape assay. DNA extracted from specimens collected during the same cruise and preserved in ethanol for approximately 40 days was used as a control. The DIN was calculated using the TapeStation analysis software based on fragmentation; its typical range is 0–10. A DIN value of 1 indicates highly degraded DNA, whereas a value of 10 represents completely intact genomic DNA.

**Figure 19 biology-14-00730-f019:**
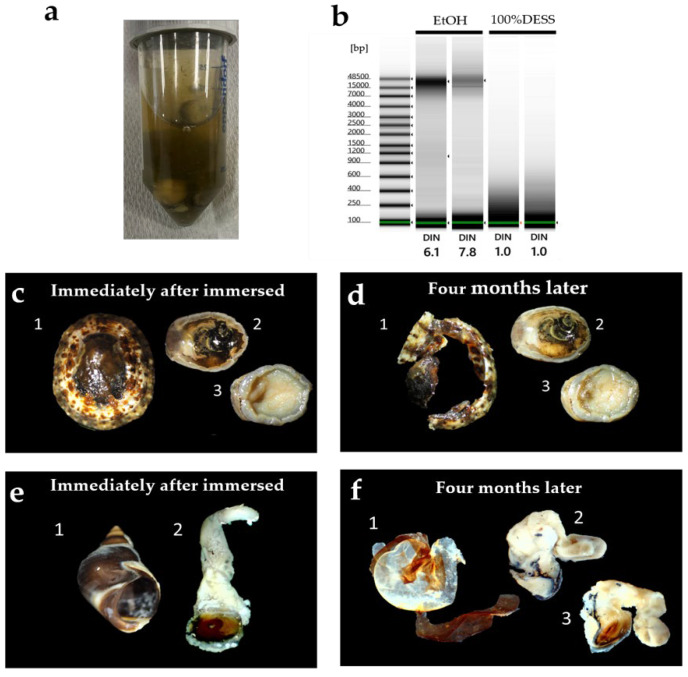
Comparison of the DNA quality and morphology extracted from tissues preserved in ethanol or DESS. Genomic DNA was analyzed using the Agilent TapeStation 4400 system and the Agilent Genomic ScreenTape assay. (**a**) Photograph of a tube containing multiple specimens of mollusks and crustaceans preserved at RT for 3 months in 20 mL DESS immediately after collection. (**b**) Electorophoresis results of DNA extracted from ethanol-preserved specimens collected during the same period and tissue samples preserved in DESS as shown in (**a**). (**c**,**d**) Intact shells and soft bodies of *Lottia kogamogai* immersed in DESS solution in dorsal view (2) and ventral view (3). (**e**,**f**) Intact shells and soft bodies of *Echinolittorina radiata* immersed in DESS solution for 2 weeks (**e**) and 4 months (**f**). Intact shells in aperture view, and soft bodies in dorsal view (**f**, 2) and ventral view (**e**, 2 and **f**, 3).

**Figure 20 biology-14-00730-f020:**
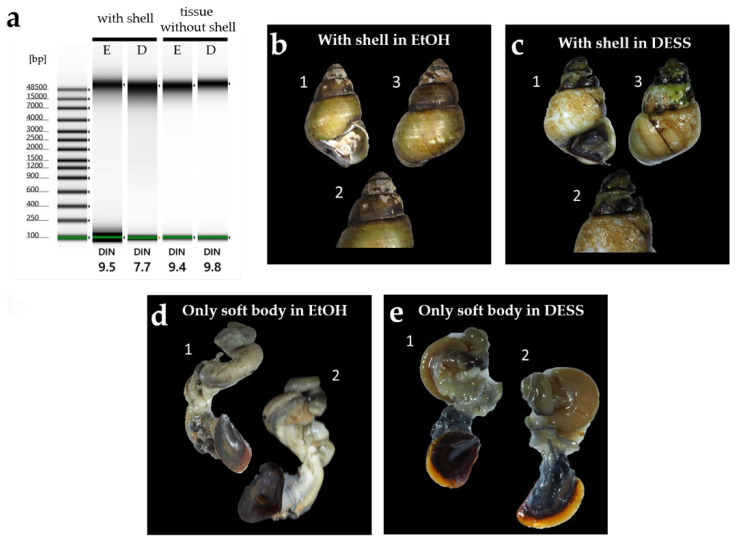
Comparison of the DNA quality and morphology extracted from *S. quadrata histrica* tissues with or without intact shells, preserved in ethanol or DESS. Genomic DNA was analyzed using the Agilent TapeStation 4400 system and the Agilent Genomic ScreenTape assay. (**a**) Electrophoresis results of DNA extracted from ethanol- and DESS-preserved specimens with and without shells sampled at the same time. (**b**,**c**) Photographs of the shells of the two specimens preserved in 100% ethanol (**b**) and in DESS (**c**). Apertural view (1), apex of the shell (2), and dorsal view (3). (**d**,**e**) Photographs of the soft bodies of the two specimens preserved in 100% ethanol (**d**) and in DESS (**e**) without shells. Ventral view (1) and dorsal view (2).

**Figure 21 biology-14-00730-f021:**
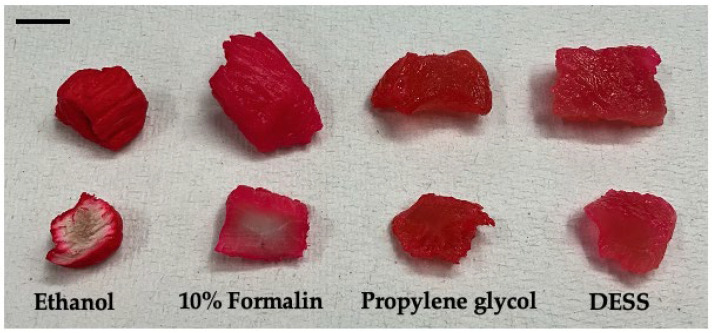
Comparison of tissue penetration by different preservative solutions. Surface (**upper**) and cross-section (**lower**) of chicken breast tissue (approximately 1 × 1 × 2 cm) after 24 h immersion in 50 mL solutions of ethanol, 10% formalin, propylene glycol, and DESS containing 100 µL of red ink. The black bar indicates 1 cm.

**Figure 22 biology-14-00730-f022:**
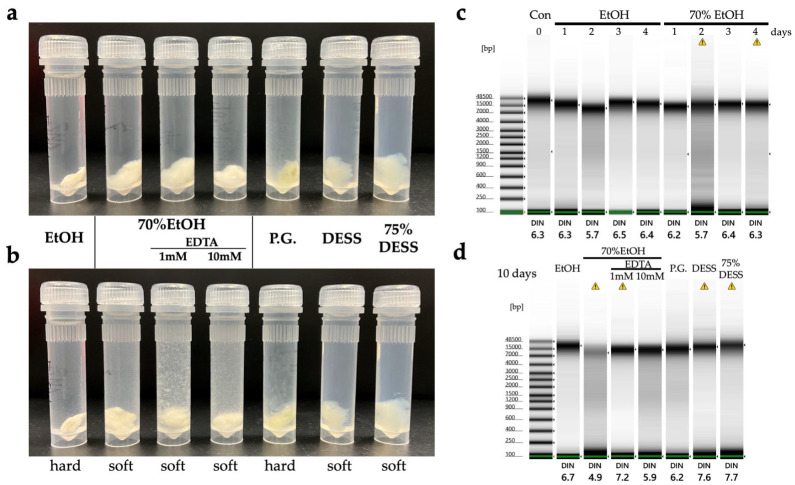
Comparison of preservation solutions for DNA degradation during a short period. (**a**,**b**) Photographs of bird muscle tissue samples (approximately 0.1–0.2 g) preserved in 1.8 mL of ethanol, 70% ethanol (without EDTA, with 1 mM or 10 mM EDTA), propylene glycol, DESS, or 75% DESS after 10 days. (**a**) Samples were under static storage conditions at 25 °C. (**b**) Samples were immediately mixed after inversion. The tissue consistency is indicated below. (**c**,**d**) Electrophoresis results of DNA extracted from samples preserved in ethanol or 70% ethanol for 0 to 4 days (**c**) or preserved in each solution for 10 days (**d**). The DIN was calculated using TapeStation analysis software based on fragmentation; its typical range is 0–10. A DIN value of 1 indicates highly degraded DNA, whereas a value of 10 represents completely intact genomic DNA. The yellow alert indicates a sample concentration that is outside the recommended range.

**Figure 23 biology-14-00730-f023:**
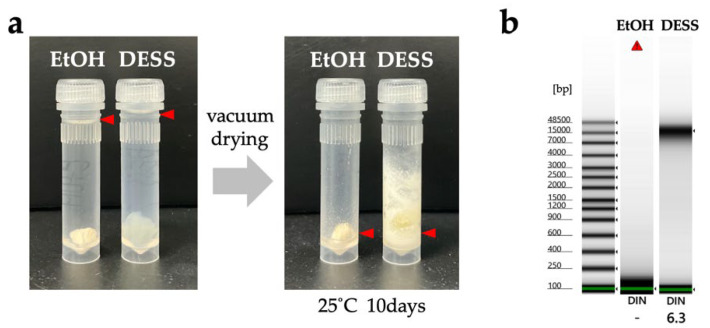
Tissue and DNA in evaporated preservative solutions. (**a**) A total of 2 mL tubes filled with ethanol and DESS with preserved muscle tissue (left). Tubes after vacuum drying and stored at 25 °C for 10 days (right). The red arrow indicates the liquid level. (**b**) DNA was extracted from a portion of the tissues shown in (**a,** right), and DNA quality was assessed using TapeStation. The red triangle indicates that the DNA preserved in ethanol was at the detection limit. The left-most lane shows the size marker.

**Figure 24 biology-14-00730-f024:**
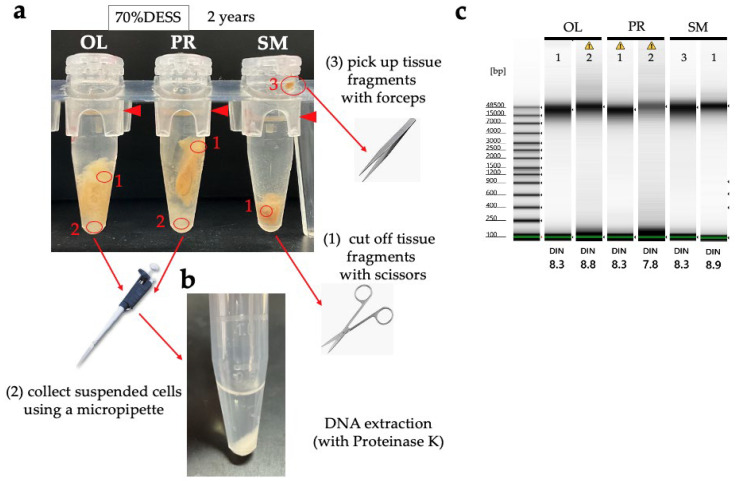
Bird muscle tissue preserved in 100% DESS at RT under dark conditions for 2 years, and its DNA. (**a**) Photograph of tubes containing bird muscle tissues preserved in 70% DESS at room temperature under dark conditions for 2 years. Muscles were obtained from the following species: OL (*Otus lettia*), PR (*Pellorneum ruficeps*), and SM (*Sturnia malabarica*). Red circles indicate the sites used for DNA extraction. Red arrows indicate the tools used for tissue collection (1: surgical scissors, 2: pipette, 3: forceps) and the resulting collected tissues. Red text 1: A tissue fragment of approximately 2 mm was cut with scissors. Red text 2: Approximately 400 µL of suspended tissue, along with DESS, was collected using a pipette and centrifuged to obtain the state shown in (**b**). Then, the supernatant was discarded, and the cell pellet was used for DNA extraction. Red text 3: Tissue fragments not immersed in DESS that adhered to the lid were removed with forceps. (**c**) Analysis results of DNA obtained from each tissue using TapeStation 4400 Genomic DNA. DIN values indicate DNA integrity, with 1 indicating the most degraded state and 10 indicating nondegraded DNA. The yellow triangle error messages indicate that the DNA concentration does not meet the recommended concentration for evaluating DNA values.

**Figure 25 biology-14-00730-f025:**
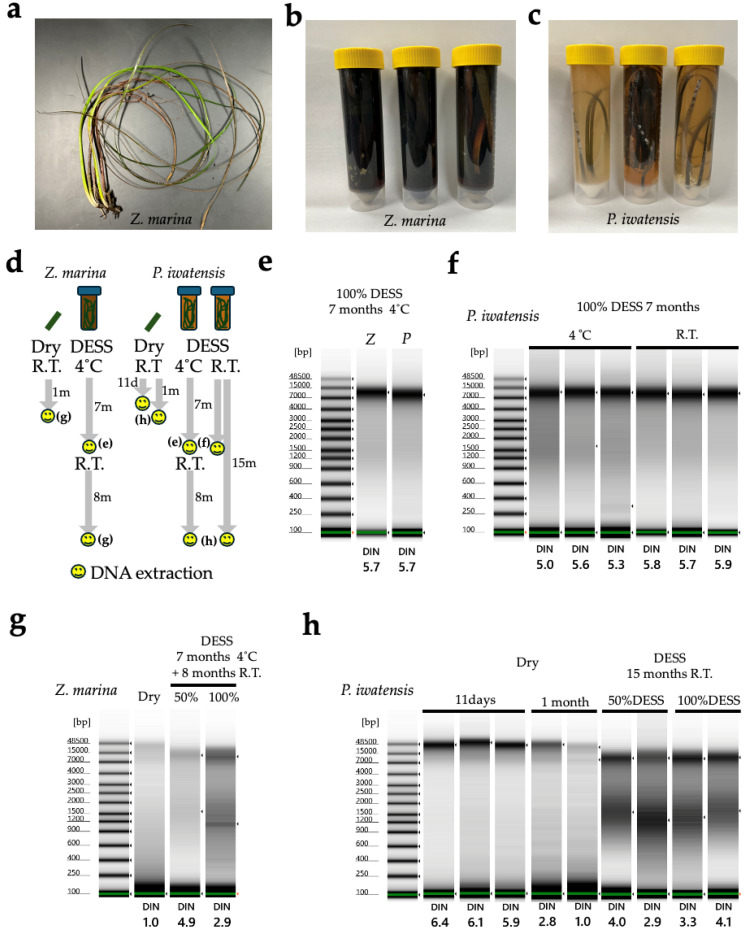
Comparison of DNA quality between samples transported to the laboratory under refrigeration, dried with silica gel, and processed immediately, versus samples preserved at room temperature in DESS for 7 or 16 months. (**a**) Photograph of *Zostera marina* after being removed, and the samples were transported to the laboratory in a cooler box immediately after collection. (**b**,**c**) Samples preserved in DESS for 7 months: *Z. marina* (**b**) and *Phyllospadix iwatensis* (**c**). The white precipitate consists of crystallized salts. (**d**) Schematic timeline of preservation methods, storage conditions, and DNA extraction points for dried leaves and DESS-preserved samples of *Z. marina* and *P. iwatensis*. (**e**–**h**) Electrophoresis results of DNA extracted from specimens preserved in DESS at 4 °C for 7 months (**e**), *P. iwatensis* preserved in DESS at 4 °C and RT for 7 months (**f**), *Z. marina* leaves dried for one month and preserved in 50% or 100% DESS at 4 °C for 7 months followed by 8 months at RT (**g**), and *P. iwatensis* leaves dried for 11 days or one month and preserved in 50% or 100% DESS at RT for 15 months (**h**). *Z* and *P* represent *Zostera marina* and *Phyllospadix iwatensis*, respectively. The DIN was calculated by the TapeStation analysis software based on fragmentation; its typical range is 0–10. A DIN value of 1 indicates highly degraded DNA, whereas a value of 10 represents completely intact genomic DNA.

**Figure 26 biology-14-00730-f026:**
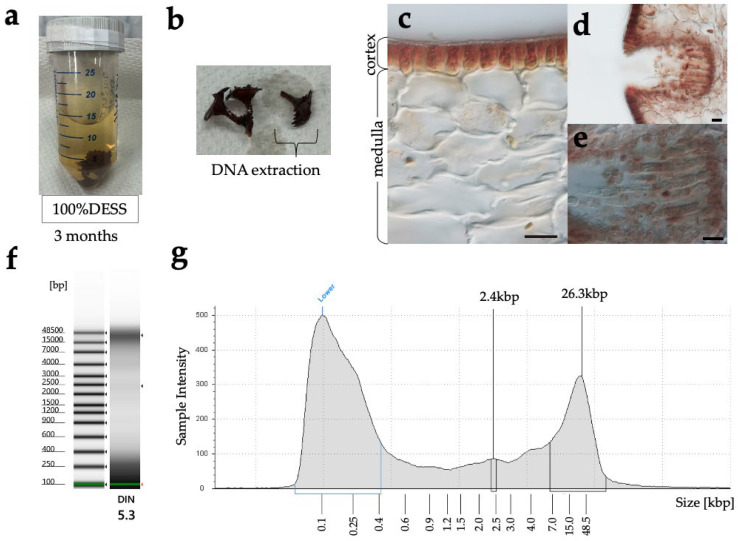
Morphology and DNA quality of the specimen preserved at RT in DESS for 3 months. Light microscopy observation of the cross-sections was performed to evaluate the morphological effects of DESS preservation. DNA quality was analyzed using the Agilent TapeStation 4400 system and the Agilent Genomic DNA ScreenTape assay to determine DIN. (**a**) *Turbinaria ornata* specimen preserved in DESS immediately after collection, and after 3 months at RT (**b**) *Turbinaria ornata* (0.1 g wet weight) removed from DESS, with a portion excised for DNA extraction. (**c**–**e**) Light micrographs of specimen cross-sections taken at 400× (**c**,**e**) and 200× (**d**) magnification. Scale bar = 20 µm. (**f**,**g**) The gel image (**f**) and electropherogram (**g**) of the extracted DNA. This sample showed a yellow alert signifying a sample concentration that was outside the recommended range. The first peak at 100 bp is a molecular weight marker standard. The DIN was calculated by the TapeStation analysis software based on fragmentation; its typical range is 0–10. A DIN value of 1 indicates highly degraded DNA, whereas a value of 10 represents completely intact genomic DNA.

**Figure 27 biology-14-00730-f027:**
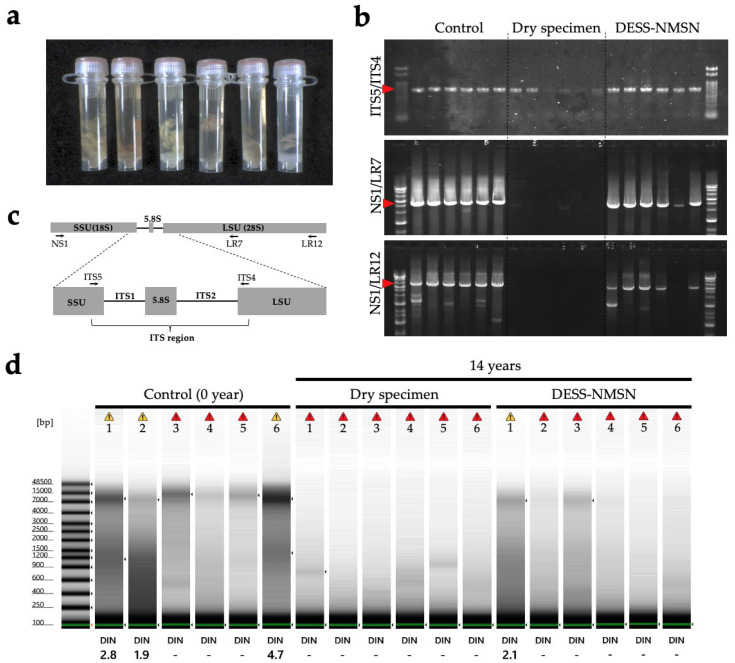
Comparison of DNA quality and PCR products extracted from dried specimens or DESS-NMNS tissue preserved for 15 years. (**a**) Fungal tissue preserved in DESS-NMNS for 15 years. (**b**) Ribosomal RNA nuclear genes and ITS regions. The small arrows observed from the rRNA and ITS regions at magnification demonstrate the annealing position of the primers used for PCR amplification. (**c**) Gel electrophoresis results of PCR products amplifying the ITS (ITS5/ITS4) and rRNA (NS1/LR7, NS1/LR12) regions using control DNA (extracted 14 years ago), DNA extracted from old dried specimens, or DESS specimens as templates. The ends are size markers, using 100 bp DNA Ladder PLUS [100 bp–3 kb] for ITS5/ITS4 and Gene Ladder Wide 1 [100 bp–20 kb] (Nippon Genetics, Tokyo, Japan) for NS1/LR7, NS1/LR12. (**d**) Quality comparison between DNA extracted from fungi after collection (0 year) and DNA extracted from dried specimens or DESS tissue specimens preserved for 14 years. The DIN was calculated by the TapeStation analysis software A0.01.05 (Agilent) based on fragmentation; its typical range is 0–10. A DIN value of 1 indicates highly degraded DNA, whereas a value of 10 represents completely intact genomic DNA. A DIN value of (-) indicates that the DNA concentration was too low to measure the DIN. The yellow alert indicates a sample concentration that is outside the recommended range. The red alert indicates DNA concentrations below the DIN value estimation limit. The first peak at 100 bp is a molecular weight marker standard.

**Figure 28 biology-14-00730-f028:**
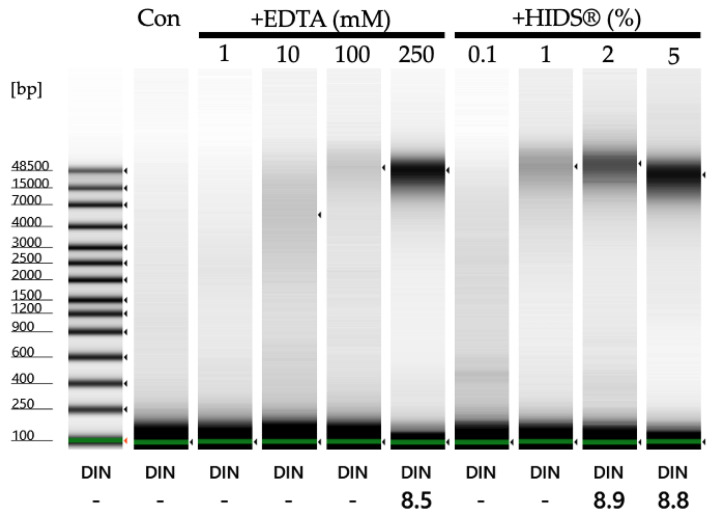
DNA degradation inhibitory effect at various concentrations of EDTA and HIDS^®^. “Con” represents the control, where only DNA was added to the DNase buffer. The DNase inhibitor addition groups, from left to right, are: EDTA at 1 mM, 10 mM, 100 mM, and 250 mM, and HIDS^®^ at 0.1%, 1%, 2%, and 5% added to the DNase buffer. After heat treatment at 37 °C for 30 min and 95 °C for 10 min, DNA was purified using beads, and DNA fragmentation was investigated using the TapeStation Genomic Tape (Agilent).

**Figure 29 biology-14-00730-f029:**
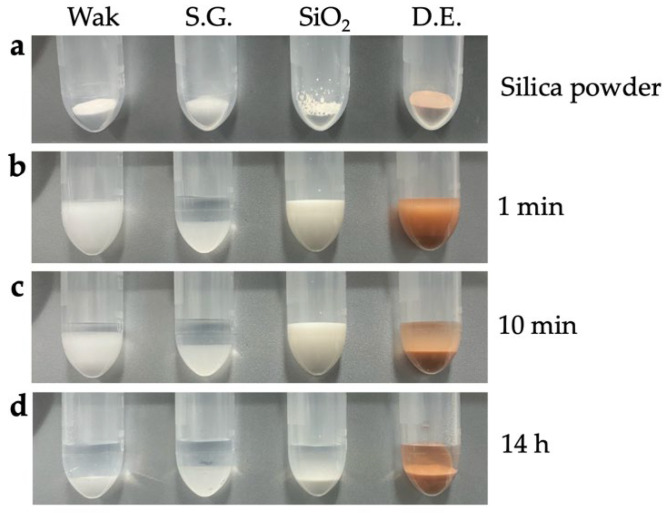
Appearance of various silica powders and properties of silica suspensions. (**a**) Silica dry powders in 2 mL tubes. Wakosil = 0.036 g, Silica gel = 0.0383 g, SiO_2_ = 0.0334 g, diatomaceous earth = 0.0329 g (**b**–**d**) After adding 10 times the weight of water to the silica in (**a**) and mixing, shown at 1 min (**b**), 10 min (**c**), and 14 h (**d**) after settling. The gray plate in the background is a magnet from a magnetic stand.

**Figure 30 biology-14-00730-f030:**
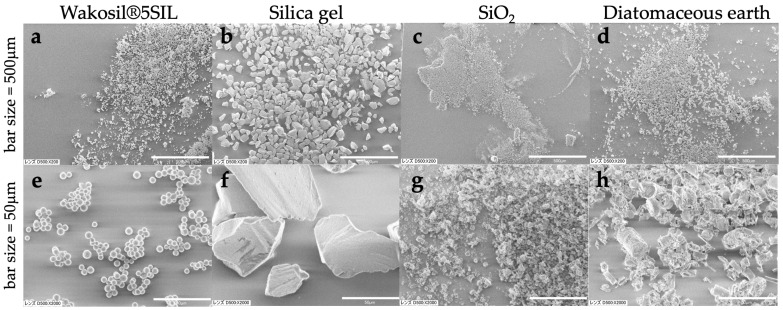
Electron microscope photographs of silica. (**a,e**) Wakosil^®^5SIL; (**b,f**) Silica gel; (**c,g**) SiO_2_; (**d,h**) Diatomaceous earth. Panels (**a**–**d**) were captured at a magnification of 500 × 200, and panels (**e**–**h**) at 500 × 2000. White scale bars indicate 500 µm in (**a**–**d**) and 50 μm in (**e**–**h**). (**a**–**h**) Each electron microscope (KEYENCE VHX-D510) photograph shows the lens D500 and magnification in the lower left corner, at 200× (**a**–**d**) and 2000× (**e**–**h**), respectively.

**Figure 31 biology-14-00730-f031:**
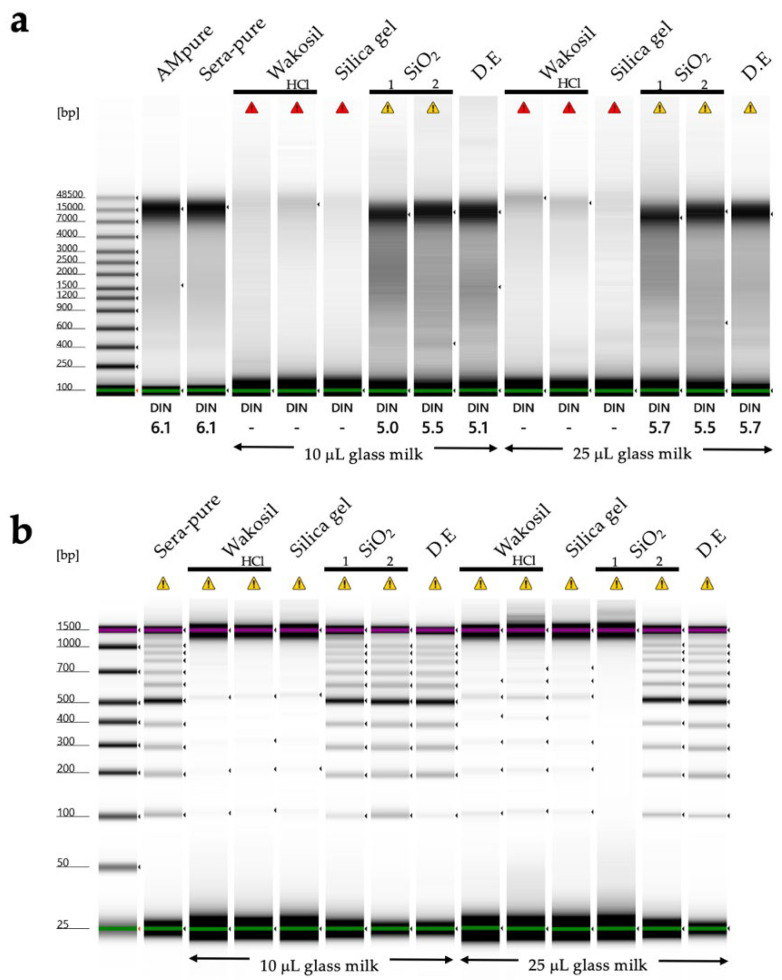
DNA purification using the Boom method and differences in silica properties. (**a**) Results of DNA purification after adding 10 μL or 25 μL of glass milk to 200 μL of gDNA solution (total 500 ng DNA) extracted from bird muscle with an equal volume of AL buffer. (**b**) Results of purifying 100 bp DNA ladder (650 ng, Nippon genetics, Japan) under similar conditions. AMpure and Sera-pure data are included as reference data for DNA purification using magnetic beads. D.E. indicates diatomaceous earth. Glass milk demotes a suspension of silica in either water or 0.01N HCl. The yellow alert indicates a sample concentration that is outside the recommended range. The red alert indicates DNA concentrations below the DIN value estimation limit. The first peak at 100 bp or 25 bp are molecular weight marker standards.

**Figure 32 biology-14-00730-f032:**
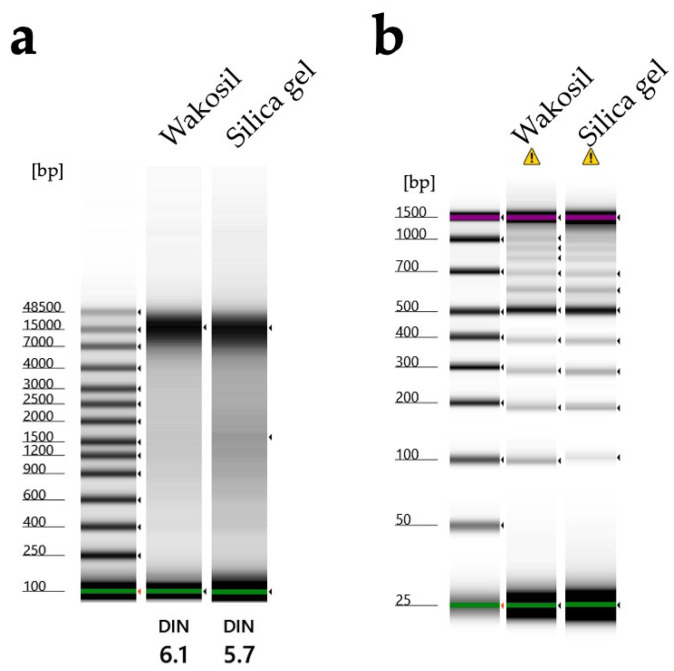
DNA purification using the Boom method with chaotropic agents and ethanol, and differences in silica properties. (**a**) Results of DNA purification after adding 10 μL of glass milk to 200 μL of gDNA solution (total 500 ng DNA) extracted from bird muscle with an equal volume of AL buffer. (**b**) Results of purifying 100 bp DNA ladder (650 ng) under similar conditions. The yellow alert indicates a sample concentration that is outside the recommended range. The first peak at 100 bp or 25 bp are molecular weight marker standards.

**Table 1 biology-14-00730-t001:** List of species and preservation solutions examined in this study.

Taxa					Preservation Condition
Animal	Invertebrates	Nematoda (free-living marine nematode) ^1^		Oncholaimida Phanodermatidae Enoplida Adoncholaimus	DESS
		Arthropoda	Insecta	Membracidae Coleoptera Scarabaeidae Chrysomelidae Cerambycidae Staphylinidae Carabidae Endomychidae Curculionidae Pyrochroidae Nitidulidae Mordellidae Cantharidae *Protaetia orientalis*	DESS
				Mantodea	Ethanol, DESS
				Orthoptera	Ethanol, DESS, DESS-NMNS ^2^
				*Prionus insularis*	Air dry, Ethanol, DESS, Ethanol + DESS,
			Spider	Okumura various (Appendix A)	70% Ethanol, DESS, etc.
			Mysida	*Neomysis japonica*	Ethanol, DESS, formalin
		Echinodermata	Holothuroidea cucumber	*Elpidia kurilensis* *Peniagone* sp. *Pseudostichopus* sp.	Ethanol, DESS
		Mollusca		*Echinolittorina radiata* *Lottia kogamogai* *Nassarius siquijorensis* *Sinotaia quadrata histrica*	100% Ethanol, DESS
	Vertebrates	Aves (Birds)	Birds	Charadriiformes Columbiformes Passeriformes Piciformes Psittaciformes Strigiformes *Gallus gallus domesticus*	Ethanol, DESS
Plant	Seagrasses			*Zostera marina* *Phyllospadix iwatensis*	DESS
Algae	Seaweeds	Ochrophyta	Brown alga	*Turbinaria ornata*	DESS
Fungi	Mushrooms	Basidiomycota		*Pluteus* sp. *Cortinarius* sp. *Russula* sp. *Entoloma* spp. *Thaxterogaster* sp.	modified DESS ^2^ (DESS-NMNS)

^1^ Shimada et al. 2012, 2017, 2024 [20,21,22]. ^2^ Hosaka and Castellano, 2008 [23].

**Table 2 biology-14-00730-t002:** Composition and reagent catalog numbers of DESS and DESS-NMNS.

Composition	DESS	DESS-NMNS ^3^	Cat. No.
DMSO	20%	20%	FUJIFILM, Tokyo, Japan, 043-07216
EDTA ^1^	250 mM	250 mM	DOJINDO, Mashiki, Japan, 345-01865
NaCl	saturated	saturated	FUJIFILM, 195-01663
Tris-HCl (pH 8.0) ^2^	-	100 mM	NACALAI TESQUE, Kyoto, Japan, 35406-75
sodium sulfite (Na_2_SO_3_)	-	100 mM	FUJIFILM, 192-03415

^1^ EDTA was adjusted the pH 8.0 using NaOH. ^2^ Tris-HCl was adjusted the pH 8.0 using HCl. ^3^ Hosaka and Castellano, 2008 [23].

**Table 3 biology-14-00730-t003:** Advantages and disadvantages of reagents used in preservation solutions and various specimen treatments for specimen and DNA preservation.

	Treatment Method	Advantages	Disadvantages
Reagent	99% ethanol ^1^	Inexpensive Safe Dehydrating ability	DNase inhibitory effect: Medium Non-shippable concentration Volarility Tissue hardening
	70% ethanol	Inexpensive Safe Shippable concentration Maintains tissue flexibility	DNase inhibitory effect: Low ^2^* Volarility ^3^*
	99.5% ethanol ^4^	Safe Dehydrating ability	DNase inhibitory effect: Medium Non-shippable concentration Volarility Tissue hardening
	Dimethyl sulfoxide (DMSO)	Cell permeability Dehydrating ability Cell cryoprotectant (5~10%) DNAphotoprotection Extremely low volatility	Low cell toxicity
	Propylene glycol (P.G.)	Safe Extremely low volatility Dehydrating ability(preservative effect >10%) Cell permeability Cell cryoprotectant (1~7%)	DNase inhibitory effect: Medium
	EDTA (pH 8.0)	DNase inhibitory effect: High (Chelating action)	Environmental impact: High (Difficult biodegradation) Morphological damage to the calcified organisms
	HIDS^® 5^	DNase inhibitory effect: High (Chelating action) Biodegradable	Environmental impact: Low Morphological damage to the calcified organisms
	NaCl	DNA stabilization Preservative effect (osmotic pressure)	-
	Tris-HCl (pH 8.0)	pH stabilization	-
	Sodium Sulfite ^6^	Antioxidant Safe	-
Treatment Method	Drying	DNase inactivation	Morphological damage/deformation DNase reactivation due to moisture absorption
	Heat (dry heat/hot water) (e.g., 55 °C 1 h, 65 °C 30 min, 90 °C 10 min)	Irreversible DNase inactivation	Equipment/hot water preparation requirement Morphological damage
	Ultrasonic cleaning ^7^	Specimen cleaning High cleaning power	DNA fragmentation ^8^ Morphological damage ^9^
	Refrigeration (4 °C)	Reduced DNase activity Morphology preservation	Equipment maintenance Power requirements
	Freezing (−80 °C)	DNase inactivation Morphology preservation in the cryoprotectant	Equipment maintenance Power requirements

^1^ Industrial ethanol. ^2^* The addition of 10 mM EDTA enhanced the DNase inhibition effect compared to anhydrous ethanol. ^3^* Even if ethanol evaporates, the DNase inhibition effect of EDTA remains. ^4^ Reagend grade ethanol. ^5^ Biodegradable chelating agent (Nippon Shokubai, Co., Ltd., Osaka, Japan). ^6^ Antioxidant. ^7^ Sometimes used for cleaning insect specimens. ^8^ Ultrasonic waves are commonly used for DNA fragmentation in solution, but the extent of DNA damage in the specimens is unknown. ^9^ Specimens may be damaged depending on the intensity of the ultrasonic waves, requiring adjustment for each species.

**Table 4 biology-14-00730-t004:** Cost comparison of the preservation solutions.

Solution Name	Cost Per 100 mL (the USD, JPY) ^1^	Cost Per 1 Tube (USD, JPY) ^2^
99% ethanol ^3^	USD0.43–0.69, JPY62–100	USD0.014, JPY2
70% ethanol ^4^	USD0.34–0.52, JPY50–75	USD0.01, JPY1.5
70% ethanol [10 mM EDTA (pH 8.0)]	USD0.62–0.83, JPY90–120	USD0.017, JPY2.4
≥99.5% special grade ethanol ^5^	USD3.1, JPY450	USD0.062, JPY9
Propylene glycol (P.G.)	USD3.44, JPY500	USD0.069, JPY10
DESS (DMSO/EDTA/Saturated Salt)	USD8.26, JPY1200	USD0.17, JPY25
DESS-NMNS (DESS + Tris + Sodium Sulfite)	USD9.85, JPY1430	USD0.20, JPY29
PE-SS (P.G./EDTA/Saturated Salt) ^6^	USD7.64, JPY1110	USD0.15, JPY22
PH-SS (P.G./HIDS^®^/Saturated Salt) ^7^	USD3.79, JPY550	USD0.076, JPY11
RNAlater (RNAprotect Reagent) ^8^	USD134.96, JPY19,600	USD1.35, JPY196

^1^ Costs are shown in Japanese JPY (¥) as of January 2025, with USD equivalents calculated using an exchange rate of JPY145.23 = USD1.00. Dollar values are based on reagent prices in Japan. ^2^ Based on 2 mL of solution per tube. ^3^ 18 L of 99% industrial-grade synthetic ethanol. ^4^ The cost includes dilution with sterilized water. ^5^ Guaranteed reagent grade. ^6^ Preservation solution prepared by replacing DMSO in DESS with propylene glycol. ^7^ Preservation solution prepared by replacing EDTA in PE-SS with 5%HIDS^®^. ^8^ QIAGEN RNAprotect Tissue Reagent (250 mL) Cat. No. 76106 (QIAGEN, Hilden, Germany).

## Data Availability

The original contributions presented in this study are included in the article and Appendix A. Further inquiries can be directed to the corresponding author.

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
