# Peer review of "DNA Specimen Preservation Using DESS and DNA Extraction in Museum Collections"

_biology, 2025, doi:10.3390/biology14060730_

Round 1
Reviewer 1 Report
Comments and Suggestions for Authors
The work raises important questions about the conservation of genetic resources and the preservation of museum collections. This aspect is usually important to protect specimens for years to come in terms of research, so that tissue damage and DNA degradation do not occur. On the other hand, in the context of research on small organisms or their fragments, the authors raise the important issue of archival sample size. The question then arises: to store further or to study? Is this sample sufficient for research? I find the issue raised in the paper interesting, also for use in normal research and tissue preparation.
However, I have a few questions for the authors.
1) What was the reason for the wide range of pH changes as a result of Zostera marina storage (line 401-402)?
2) Have you considered the effect of tissue and organism hydration on ethanol and DESS evaporation rates?
3) In the other organisms tested, was there also no curing effect of DESS as in the case of shrimp?
4) What do the authors mean by 'leaf section' (line 1321)? Please clarify.
5) In subsection 3.3.1, I suggest that the cost of reagents be stated in a more universal currency (e.g. $), which will allow a better understanding of the financial issues and limitations of the different methods.
6) And in the case of DNA extraction from museum specimens, have forensic DNA isolation and extraction techniques been used? Typically, these techniques require a trace sample for DNA isolation.
Author Response
Comments and Suggestions for Authors:
The work raises important questions about the conservation of genetic resources and the preservation of museum collections. This aspect is usually important to protect specimens for years to come in terms of research, so that tissue damage and DNA degradation do not occur. On the other hand, in the context of research on small organisms or their fragments, the authors raise the important issue of archival sample size. The question then arises: to store further or to study? Is this sample sufficient for research? I find the issue raised in the paper interesting, also for use in normal research and tissue preparation.
However, I have a few questions for the authors.
1) What was the reason for the wide range of pH changes as a result of Zostera marina storage (line 401-402)?
Author’s response:As reported in Maeda et al. 1996 (https://www.jstage.jst.go.jp/article/jplantres1887/79/938/79_938_422/_pdf/-char/en), Z. marina contains sulfates, which likely dissolved into the DESS solution, causing the pH to decrease. We have added the reason in Line 421-422.
2) Have you considered the effect of tissue and organism hydration on ethanol and DESS evaporation rates?
Author’s response:In this study, we conducted evaporation experiments without including tissues or organisms in the ethanol or DESS, in order to measure pure evaporation rates. We did not examine the effect of tissue and organism hydration, but since the relative amount is small, we believe it has minimal impact on evaporation rates.
3) In the other organisms tested, was there also no curing effect of DESS as in the case of shrimp?
Author’s response:No curing effect from DESS was observed in any of the species tested.
4) What do the authors mean by 'leaf section' (line 1321)? Please clarify.
Author’s response:Thank you for pointing this out; we agree the “leaf section” does not clearly convey the meaning. We have changed it to “leaf position and maturity”.
5) In subsection 3.3.1, I suggest that the cost of reagents be stated in a more universal currency (e.g. $), which will allow a better understanding of the financial issues and limitations of the different methods.
Author’s response:Indeed, stating costs in a more universal currency would make it more understandable to most readers. Since reagent prices in Japan differ from those in other countries, we cannot calculate them accurately. Therefore, we have included both the reagent prices in Japan (yen) and their converted dollar values in Table 4.
6) And in the case of DNA extraction from museum specimens, have forensic DNA isolation and extraction techniques been used? Typically, these techniques require a trace sample for DNA isolation.
Author’s response:The forensic DNA extraction techniques and our method share the same fundamental principles. In forensic DNA extraction, it is necessary to minimize the risk of human genome contamination to the greatest extent possible, so reagent manufacturer kits products under strict conditions should be used. However, for non-human museum specimens, such stringent measures are not required (although filter tips must always be used).
Reviewer 2 Report
Comments and Suggestions for Authors
Manuscript «DNA Specimen Preservation using DESS and DNA Extraction in Museum Collections: A Case Study Report» by Eri Ogiso-Tanaka with co-authors provides a detailed report on the methodology of DNA preservation. The authors elaborated and comprehensively analysed application of a solution containing DMSO, EDTA and NaCl for storing DNA in specimens. The results of this study are quite convincing, and I have no major comments. The introduction section gives the reader a clear understanding of the problem. The Materials and methods section is rather detailed. The Results and Discussion is quite lengthy however I have no idea what part could be ommited or restricted. For brief result the reader can refer to the Conclusion section, not to an abstract. I don't think that Abstract should be rewritten however the Simple Summary can be modified. I suppose that the authors can use Simple Summary as some sort of highlights or a quick guide. Just express recommendations for DNA preservation obtained in this study. Start with „Here we investigated the effectiveness of traditional preservation methods (EtOH, EtOH+EDTA…) versus DESS (DMSO/… Best results…“.
Author Response
Comments and Suggestions for Authors
Manuscript «DNA Specimen Preservation using DESS and DNA Extraction in Museum Collections: A Case Study Report» by Eri Ogiso-Tanaka with co-authors provides a detailed report on the methodology of DNA preservation. The authors elaborated and comprehensively analysed application of a solution containing DMSO, EDTA and NaCl for storing DNA in specimens. The results of this study are quite convincing, and I have no major comments. The introduction section gives the reader a clear understanding of the problem. The Materials and methods section is rather detailed. The Results and Discussion is quite lengthy however I have no idea what part could be ommited or restricted. For brief result the reader can refer to the Conclusion section, not to an abstract. I don't think that Abstract should be rewritten however the Simple Summary can be modified. I suppose that the authors can use Simple Summary as some sort of highlights or a quick guide. Just express recommendations for DNA preservation obtained in this study. Start with „Here we investigated the effectiveness of traditional preservation methods (EtOH, EtOH+EDTA…) versus DESS (DMSO/… Best results…“.
Author’s response:Thank you for your advice. Based on the comments, we have modified the Simple Summary as follows: “Here we investigated the effectiveness of traditional preservation methods (such as ethanol) versus DESS (DMSO/EDTA/saturated NaCl solution) with regards to maintaining both morphology and DNA integrity using our museum specimens. Our results demonstrated that preserving tissues in DESS maintained high molecular weight DNA across all examined species. When preserving whole organisms, the optimal preservation solution conditions for maintaining both morphological features and DNA integrity varied among species, but DESS was effective for preserving both morphology and DNA except for species with calcium carbonate skeletons and shells. These findings suggest that DESS utilization for specimen preservation is effective across many species, not only for long-term storage in environments without freezer facilities but also for temporary preservation until freezing is possible."